# Extrinsic hydrophobicity-controlled silver nanoparticles as efficient and stable catalysts for CO$_2$ electrolysis

Young-Jin Ko [1,10], Chulwan Lim[1,2,10], Junyoung Jin[3,4,10], Min Gyu Kim [5], Ji Yeong Lee [6], Tae-Yeon Seong[4], Kwan-Young Lee [2], Byoung Koun Min [1], Jae-Young Choi[7,8], Taegeun Noh[9], Gyu Weon Hwang [3]✉, Woong Hee Lee [1]✉ & Hyung-Suk Oh [1,7,8]✉

To realize economically feasible electrochemical CO$_2$ conversion, achieving a high partial current density for value-added products is particularly vital. However, acceleration of the hydrogen evolution reaction due to cathode flooding in a high-current-density region makes this challenging. Herein, we find that partially ligand-derived Ag nanoparticles (Ag-NPs) could prevent electrolyte flooding while maintaining catalytic activity for CO$_2$ electroreduction. This results in a high Faradaic efficiency for CO (>90%) and high partial current density (298.39 mA cm$^{-2}$), even under harsh stability test conditions (3.4 V). The suppressed splitting/detachment of Ag particles, due to the lipid ligand, enhance the uniform hydrophobicity retention of the Ag-NP electrode at high cathodic overpotentials and prevent flooding and current fluctuations. The mass transfer of gaseous CO$_2$ is maintained in the catalytic region of several hundred nanometers, with the smooth formation of a triple phase boundary, which facilitate the occurrence of CO$_2$RR instead of HER. We analyze catalyst degradation and cathode flooding during CO$_2$ electrolysis through identical-location transmission electron microscopy and *operando* synchrotron-based X-ray computed tomography. This study develops an efficient strategy for designing active and durable electrocatalysts for CO$_2$ electrolysis.

The electrochemical reduction of carbon dioxide has great potential for the storage of renewable electricity in carbon-based fuels and value-added chemicals[1–3]. However, the electrochemical CO$_2$ reduction reaction (CO$_2$RR) is not yet economically feasible because of its low energy efficiency and low long-term durability at high current densities[4,5]. Furthermore, the competing hydrogen evolution reaction (HER), which is significantly influenced by water and ion management in a bicarbonate-based electrolyte, considerably limits the practical

[1]Clean Energy Research Center, Korea Institute of Science and Technology (KIST), Hwarang-ro 14-gil 5, Seongbuk-gu, Seoul 02792, Republic of Korea. [2]Department of Chemical and Biological Engineering, Korea University, 145, Anam-ro, Seongbuk-gu, Seoul 02841, Republic of Korea. [3]Center for Neuro-morphic Engineering, Korea Institute of Science and Technology (KIST), Hwarang-ro 14-gil 5, Seongbuk-gu, Seoul 02792, Republic of Korea. [4]Department of Materials Science and Engineering, Korea University, 145, Anam-ro, Seoul 02841, Republic of Korea. [5]Beamline Research Division, Pohang Accelerator Laboratory (PAL), Pohang 37673, Republic of Korea. [6]Advanced Analysis Center, Korea Institute of Science and Technology (KIST), Hwarang-ro 14-gil 5, Seongbuk-gu, Seoul 02792, Republic of Korea. [7]School of Advanced Materials Science & Engineering, Sungkyunkwan University (SKKU), Suwon 16419, Republic of Korea. [8]KIST-SKKU Carbon-Neutral Research Center, Sungkyunkwan University (SKKU), Suwon 16419, Republic of Korea. [9]Platform Technology Research Center, LG Chem Ltd., 30, Magokjungang 10-ro, Gangseo-gu, Seoul 07796, Republic of Korea. [10]These authors contributed equally: Young-Jin Ko, Chulwan Lim, Junyoung Jin. ✉e-mail: gwhwang@kist.re.kr; abcabac@kist.re.kr; hyung-suk.oh@kist.re.kr

applicability of the $CO_2RR$[6]. Recent studies have attempted to elucidate the influence of the intrinsic properties of catalysts on the kinetics and end-product selectivity of the $CO_2RR$ in competition with the HER[7–9]. Electrode flooding or proton/potassium ion balance, which is classified as an extrinsic property, significantly influences $CO_2RR$ selectivity. Thus, these factors should be considered when synthesizing catalysts. Electrode flooding, in particular, is deemed as a representative cause for the low Faradaic efficiency for CO ($FE_{CO}$), resulting in acceleration of the HER[10–16]; therefore, it is essential to develop an electrode with a hydrophobic surface. However, even with the abovementioned major issues, research in this field has been limited to the microstructural control of electrodes or the introduction of polytetrafluoroethylene (PTFE)[17,18]; few electrocatalysts with hydrophobic properties have been reported. Several studies investigated the influence of hydrophobic ligands on $CO_2RR$[19,20]. However, real-time observation of the relationship between the behavior of the electrolyte and the $CO_2RR$ properties in terms of hydrophobicity at the three-phase boundary of a device has not yet been reported.

Herein, we report a silver nanoparticle (Ag-NP) partially functionalized by a hydrophobic ligand that stably maintains effective CO selectivity at a high cathodic overpotential for the $CO_2RR$. Furthermore, the lipid ligand inhibits Ostwald ripening and sintering of the electrocatalyst during the $CO_2RR$, thereby facilitating maintenance of the fine nanoparticle size[21,22]. Thus, the fabricated Ag-NP electrode maintains selectivity for CO production at high current densities by using a neutral electrolyte in a zero-gap electrolyzer system. Additionally, in-situ/*operando* synchrotron-based X-ray analyses are conducted to investigate the influence of water management on the selectivity and durability of the Ag-NP catalyst. The modulated NP-ligand structure can be used with multiple metals, with various vapor−liquid−solid interfaces forming on the catalyst surface, thereby affording greater durability and product selectivity during the $CO_2RR$ than during HER.

## Results

### Structural characterization of Ag-NPs

The synthesis was based on a modified version of a previously reported method, as shown in Supplementary Fig. 1. The Ag-NPs were synthesized using a heating-up synthesis method[23]. Subsequently, tetramethylammonium hydroxide (TMAH) treatment for 2 h was conducted to partially remove the native ligands on the Ag-NPs to ensure the formation of the catalytically active area (Supplementary Fig. 2: Schematic of the synthesis process of Ag-NP). The change in the electrochemically active surface area (ECSA) with and without TMAH treatment was analyzed for the Ag/Ag$^+$ oxidation reaction (Supplementary Fig. 3). According to the Randle-Sevcik equation, the peak current is proportional to the ECSA. The ECSA of the Ag-NPs increased 3-fold after TMAH treatment for 2 h. The Ag-NP catalyst was treated with TMAH for 5 h to eliminate the maximum amount of ligand, and its original morphology was not maintained (Supplementary Fig. 4). To analyze the hydrophobic properties while maintaining the morphology of the catalyst, by leaving a portion of the ligand, physical property analysis was performed using Ag-NPs treated with TMAH for 2 h. Figure 1a and Supplementary Fig. 5 show the HR-TEM images of the Ag-NP; the corresponding particle size histograms (Fig. 1b: box plot and Supplementary Fig. 6: HR-TEM images for calculation of the particle size distribution; the image in Fig. 1a was used to estimate the particle size distribution) show the size of the Ag-NPs without the ligand layer. The HR-TEM images indicated that the size and shape of the Ag-NPs were not significantly affected by reduction of the native ligands (Supplementary Fig. 7: HR-TEM image of pristine Ag-NPs).

The characteristic lattice fringes shown in the HR-TEM images (Supplementary Fig. 5) confirmed that all Ag-NPs were icosahedrons. An icosahedron has three types of high-symmetry axes: two-, three-, and five-fold. Face-centered cubic (*fcc*) icosahedral nanocrystals are catalytically important because their exterior surface is bound by (111) planes[24,25]; however, the preponderance of corners and edges enhances the HER catalytic activity[26,27]. To minimize the surface energy, the amidogen groups of the oleylamine ligands were attached to low-coordinated sites, especially at the corner sites[28]. When the ligands coordinated on the Ag-NP surface, they were more likely to occupy the corner sites. Therefore, the $CO_2RR$ selectivity was expected to improve upon the attachment of the ligands to the corner sites of the nanocrystals.

To identify the ligands remaining on the partially ligand-bound Ag-NPs, atomic probe tomography (APT) was performed. The APT profiles of the Ag-NPs indicated the 3D distribution and concentration of ligands on the Ag-NP surfaces. For APT, the Ag-NPs were spin-coated on an e-beam-deposited Cr thin film; subsequently, a needle-shaped tip was used for focused ion beam (FIB) milling. Figure 1c, d shows the needle-shaped tip and the reconstructed 3D atom map containing several Ag-NPs and an iso-concentration surface of carbon. A thin slice (thickness = 20 nm; viewed along the z-axis) was used to analyze the projection of the Ag-NPs (Fig. 1d). C and N atoms were segregated on the Ag-NP surface. Figure 1e shows the measured atomic concentrations of the isolated Ag-NPs; N atoms were predominantly observed only on the surface of the Ag-NPs. Since the N signal might be due to residual TMAH, $^1H$ nuclear magnetic resonance (NMR) analysis was additionally performed (Supplementary Fig. 8). In the spectrum of the Ag-NPs, peaks were observed near 5.25 ppm and 2.5 ppm, which are attributed to the −CH = CH- and −CH$_2$NH$_2$ moieties of oleylamine, respectively[29]. Thus, the native ligand was visually confirmed on the surface of the Ag-NPs through APT but not TEM.

### $CO_2$-to-CO conversion performance of Ag-NPs

The electrocatalytic activity and stability of the prepared Ag-NPs for the $CO_2RR$ were investigated; the results are shown in Fig. 2. Prior to the electrochemical measurements, the Ag-NP electrocatalysts were oxidized to obtain an oxide-derived electrode for the removal of impurities on the Ag-NP surfaces. Fourier transform infrared (FT-IR) spectroscopy was conducted after oxidation to observe whether the lipid ligand changed due to the preoxidation process. For measurement under ideal conditions without the influence of the substrate (Supplementary Fig. 9: FT-IR spectrum of GDL), preoxidation was performed in a half-cell system using a double-polished silicon substrate. Despite the varying reaction environments between the zero-gap electrolyzer and the aqueous electrolyte, the variations in the ligand induced by TMAH treatment and preoxidation were observed to a certain extent through FT-IR analysis. Although the electrode was oxide-derived, the lipid ligand was not removed from the electrode surface, which indirectly suggests a slight structural alteration in the Ag-NPs. As shown in the FT-IR spectra (Supplementary Fig. 10), the peaks in the carbon–hydrogen (C−H) stretching region indicate the presence of OLA on the catalyst surface even after electrochemical treatment. A homemade zero-gap electrolyzer using gaseous $CO_2$ to accelerate the $CO_2RR$ while minimizing mass transfer resistance was used to analyze the electrochemical $CO_2RR$ performance of the Ag-NPs[30–32]. A detailed schematic of the zero-gap $CO_2$ electrolyzer system is shown in Supplementary Figs. 11 and 12. We performed a $CO_2$ electrolysis experiments by supplying 0.1 M $KHCO_3$ solution to the anode side and inserting 200 sccm of humidified $CO_2$ gas at 50°C into the cathode side. First, the $CO_2RR$ performance was determined according to the TMAH treatment time (Supplementary Fig. 13). The Ag-NP catalyst subjected to 2 h of TMAH treatment showed the highest performance because an appropriate amount of ligand was attached on the surface. The TMAH treatment for 2 h was predicted as the optimal condition to maintain the icosahedral shape (Fig. 1d and Supplementary Fig. 5) while increasing the number of active sites of Ag through the appropriate amount of ligand removal.

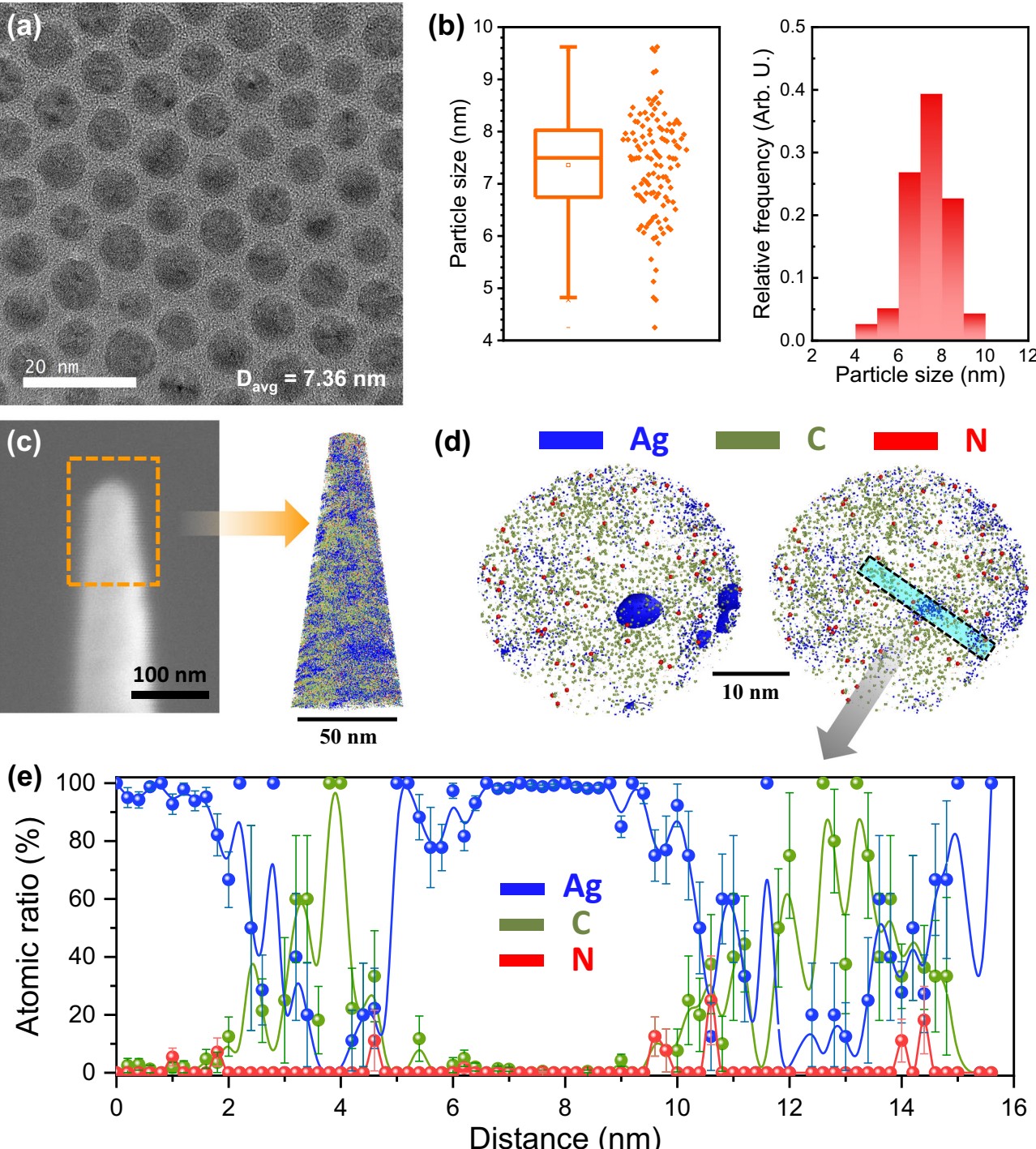

**Fig. 1 | Physical properties of the Ag-NP catalyst treated with tetra-methylammonium hydroxide (TMAH) for 2 h. a** High-resolution transmission electron microscopy (HR-TEM) image of the synthesized Ag-NP catalyst (Supplementary Fig. 1: low-magnification TEM image). **b** Box and histogram plots show the particle size distributions for the Ag-NP catalyst ($D_{avg}$ = 7.36 nm, σ = 13.2%). **c** Scanning electron microscopy image of the atomic probe tomography (APT) specimen and its 3D atom map for the Ag-NP catalyst. **d** Slices viewed along with the Ag-NP catalyst and its (**e**) atomic line profile. Left side of Fig. 1d presents a slice marked with high contrast of Ag to display the location of Ag particles, and the right side of Fig. 1d presents a slice with the contrast of the left slice adjusted equally for each element.

Therefore, the Ag-NP catalyst treated for 2 h was used to compare the performance of other Ag catalysts. The Ag-NP catalysts were sprayed onto a GDL, and gaseous $CO_2$ was supplied to the cathode side; a 0.1 M $KHCO_3$ electrolyte was circulated in the anode flow channels, which were physically separated from the Ag-NP electrode using an anion exchange membrane (AEM). Although the Ag black catalyst exhibited a higher partial current density for CO at

low cell voltages (Fig. 2a), the Ag-NP catalyst displayed a significantly higher $FE_{CO}$ value than the Ag black catalyst at high cell voltages (Supplementary Fig. 14). In particular, the Faradaic efficiency of Ag-NP catalyst exceeded 90% at a cell voltage of 3.4 V, whereas the $FE_{CO}$ of the Ag black catalyst was reduced to 75% at 3.4 V. Therefore, the CO partial current density of the Ag-NP and Ag black catalysts was turned around at a high cathodic overpotential.

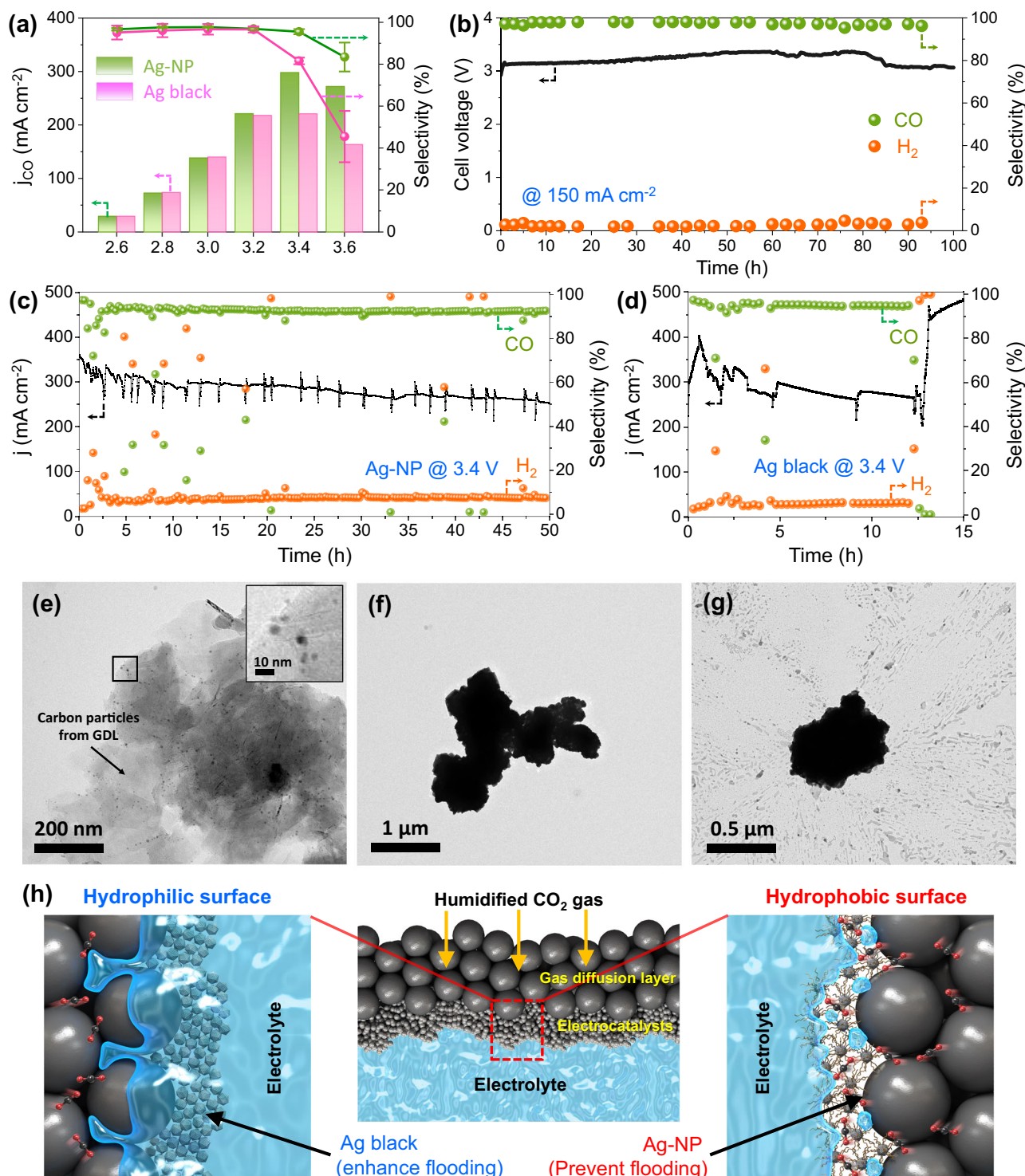

**Fig. 2 | Single-cell performance of the Ag-NP catalyst. a** Selectivity of CO and (**b**) CO partial current density versus applied cell voltage in a zero-gap $CO_2$ electrolyzer for the Ag black and Ag-NP catalysts. Ag black catalyst is a commercial fine powder of metallic Ag. Each experiment was measured three times to obtain the average value and error. **b**, **c** Durability test results for the Ag-NP catalyst in the zero-gap $CO_2$ electrolyzer at (**b**) 150 mA cm⁻² for 100 h and (**c**) 3.4 V for 50 h. **d** Durability test results for the Ag black catalyst in the zero-gap $CO_2$ electrolyzer at 3.4 V for 15 h. Selectivity of CO and $H_2$ measured during the durability tests. The electrodes of the $CO_2$ electrolyzer were prepared with 0.3 mg cm⁻² of Ag catalysts on a 10-cm² gas diffusion layer (GDL) on the cathode side. **e**–**g** Low-magnification TEM image of the (**e**) Ag-NP, (**f**) Ag black, (**g**) and Ag-PTFE catalysts after the durability test at 3.4 V. **h** Schematics (low and high magnifications) of the triple phase boundary for the hydrophilic Ag black and hydrophobic Ag-NP catalysts in the $CO_2$ electrolyzer.

Furthermore, the maximum partial current density of the Ag-NP catalyst (Ag-NP = 298.39 mA cm⁻² at 3.4 V) was almost 35% higher than that of the Ag black catalyst (221.22 mA cm⁻² at 3.4 V). These results indicate an extrinsic effect of the lipid ligand on the maintenance of the $FE_{CO}$ of the Ag-NP catalyst. In addition, the $CO_2RR$

performance of Ag-NP catalysts with various particle sizes was confirmed. As shown in Supplementary Fig. 15, the Ag-NP catalyst with an average particle size of 7.36 nm exhibited the highest current density. This is consistent with the results of a previous study indicating that Ag-NPs with a moderate size, between 5 and 10 nm,

exhibited the highest CO$_2$RR activity[33]. According to the results for CO$_2$RR performance, the Ag-NP catalyst with an average particle size of 7.36 nm was used for various analyses discussed below.

Another method used to fabricate the hydrophobic electrode was to blend in a hydrophobic material such as PTFE. For comparison with the partially ligand-derived Ag-NPs, the Ag black catalyst and PTFE were physically blended to determine the CO$_2$RR performance. As shown in Supplementary Fig. 16, the CO$_2$RR performance, according to the cell voltage, of the Ag-PTFE catalyst was similar to that of the Ag-NP catalyst. However, the total current density and CO partial current density of the Ag-PTFE catalyst at a high cell voltage were significantly lower than those of the Ag-NP catalyst. This is because of the reduction in the active surface area owing to the suppression of the flooding phenomenon at a high cell voltage (Supplementary Fig. 17: CV in the non-Faradaic potential range for ECSA measurements). In addition, there was an increase in the ohmic resistance due to PTFE (Supplementary Fig. 18: Impedance spectra at a cell voltage of −3 V). As the amount of PTFE increased, the ohmic resistance increased and affected the total current density.

A long-term durability test was conducted under chronopotentiometric conditions at a constant current of 1.5 A to confirm the feasibility of the Ag-NP catalyst for sustainable CO$_2$-to-CO conversion. As displayed in Fig. 2b, the cell voltage of the Ag-NPs was -3.0 V for 100 h, and Se$_{CO}$ remained stable (over 95% with negligible fluctuation) during the experiment. Durability tests at 3.4 V were conducted to confirm CO$_2$ starvation induced by water flooding (Fig. 2c, d). As shown in Fig. 2d, the Ag black catalyst exhibited frequent current fluctuations, significantly reducing its Se$_{CO}$ and activity. After 12 h of operation, the CO$_2$ supply ceased, and the device stopped operating. In contrast, the Ag-NP catalyst showed fewer fluctuations, with an insignificant reduction in performance. Repeated water flooding occurred; however, the current recovered immediately upon water removal before complete flooding. The lipid-ligand-derived hydrophobic surface suppressed water flooding, accelerating the transport of gaseous CO$_2$. Thus, the CO$_2$RR activity improved upon reducing the current fluctuations. In agreement with the previously mentioned performance comparison results, the Ag-PTFE catalyst exhibited durability at a high cathodic overpotential (Supplementary Fig. 19). Although the electrolyte discharged well, the current fluctuation was severe, and there was a slightly greater decrease in performance in relation to that of the Ag-NP catalyst. This is presumably due to the difference in the hydrophobicity homogeneity between the lipid ligand directly attached to Ag and the PTFE randomly distributed around Ag (Supplementary Fig. 20: EDS mapping for Ag-NP and Ag-PTFE electrodes to confirm the distribution of the lipid ligand and PTFE, respectively). The voltage or current fluctuations in zero-gap electrochemical devices have a detrimental effect on long-term performance degradation[34,35]. Therefore, the management of current fluctuation is an important part of electrochemical device research. After the durability test, the Ag black and Ag-PTFE catalysts exhibited relatively severe particle agglomeration compared to the Ag-NP catalyst (Fig. 2e–g and Supplementary Figs. 21–23). This is apparently a result of the degree of flooding and will be discussed in more detail in the subsequent section.

A schematic of the electrode surface at high overpotential for hydrophobic Ag-NP and hydrophilic Ag black catalysts is depicted in Fig. 2h. Flooding did not occur on the hydrophobic surface, and CO$_2$ gas was supplied smoothly, resulting in the formation of a triple-phase boundary. In contrast, regarding the hydrophilic surface, the flooding occurred and blocked the supply of CO$_2$, thereby obstructing the formation of a triple-phase boundary.

### Origin of the excellent performance: intrinsic/extrinsic effects due to lipid ligands

The high partial current density of CO at high cathodic overpotentials is a major advantage of the Ag-NP catalyst; to elucidate the

origin of this phenomenon, the extrinsic properties of the catalyst were analyzed. IL-TEM was conducted before and after the CO$_2$RR (Fig. 3a–d) to investigate the morphological changes in the Ag-NP catalyst during the CO$_2$RR. The reaction was conducted at −1.0 V vs. RHE (−1.610 V vs. Ag/AgCl), near the partial current density of 100 mA cm$^{-2}$ in a zero-gap electrolyzer experiment (Supplementary Table 1: the cathode potential in a zero-gap electrolyzer experiment with a Ag black catalyst obtained by adopting a reference electrode). As several bubbles were generated at a high current density, the morphological variations were observed after a long period (4 h) at a relatively low current density. Despite the slightly different reaction environments between the zero-gap electrolyzer and the aqueous electrolyte, the effect of the CO$_2$RR potential on the morphology of the catalysts was observed to a certain extent through the IL-TEM experiment. For the Ag black catalyst, the number of existing Ag clusters decreased during the reaction, with numerous Ag nanoparticles located around the original Ag clusters (Fig. 3a, b). This phenomenon, defined as cathodic corrosion, possibly occurred via the splitting/detachment (red circles) and aggregation (yellow circles) of the existing Ag clusters[36,37]. In contrast, the splitting or detachment of Ag atoms was negligible during the Ag-NP-catalyzed CO$_2$RR (Fig. 3c, d). In the single-cell experiment, the difference in the morphological changes between the Ag catalysts during the CO$_2$RR was more pronounced (Supplementary Figs. 21–23). Because the agglomeration of Ag was severe during the CO$_2$RR, the Ag black and Ag-PTFE catalysts formed Ag microparticles. However, the Ag-NP catalyst did not undergo considerable agglomeration and maintained its nanoparticle morphology. The degree of morphological changes experienced by the Ag catalysts varied slightly because of the difference in current density between the zero-gap electrolyzer and the aqueous electrolyte, but the trend of change induced by the CO$_2$RR potential was equal for the zero-gap electrolyzer and the aqueous electrolyte. This difference could be attributed to the remaining lipid ligands that were attached to the corner site of the nanoparticles. They exposed the high-crystallinity (111) plane, inducing excellent catalytic activity toward the CO$_2$RR under cation-assisted conditions[38,39] and increased the CO$_2$RR activity without cathodic corrosion. As shown in Supplementary Fig. 24, there was no change in crystallinity or particle size for Ag (111) after the CO$_2$RR. In addition, the Ag (110) surface, which is known to exhibit high activity for the CO$_2$RR, also participated in the CO$_2$RR to increase activity, and there was no degradation after the CO$_2$RR, similar to the Ag (111) surface. Although the lipid ligand could be removed by a strong base such as TMAH or a strong reducing agent such as NaBH$_4$[19], it did not create a harsh environment in which the ligand could be degraded in the zero-gap CO$_2$ electrolyzer near the neutral electrolyte.

In-situ/*operando* XANES spectroscopy at the Ag k-edge for the Ag black and Ag-NP catalysts under gaseous CO$_2$RR conditions in a homemade electrochemical device with a GDL was performed to identify the oxidation states of the Ag catalysts[40]. The penetration depth of XANES analysis is approximately 3 μm for Ag[41]. Therefore, bulk signals are detected together with surface information. However, nanoparticles can be used for surface analysis because of the low interference of the bulk signal. The ex-situ Ag k-edge XANES spectra indicated that both catalysts were predominantly in the zero-valent oxidation state (Fig. 3e). The in-situ/*operando* XANES spectrum of the Ag black catalyst exhibited a considerable anodic-potential-derived energy shift, indicating the presence of a large fraction of oxidized Ag upon fitting (Fig. 3f), which rapidly reverted to the metallic Ag phase during the CO$_2$RR. Although the spectrum of the Ag-NP catalyst exhibited a similar energy shift under anodic potential, oxidized Ag did not completely revert to the metallic Ag phase during the CO$_2$RR. The LCF results confirmed this trend (Fig. 3g). The redox reaction of Ag$^0$/Ag$_x$O$_y$, owing to an increase in the local pH, accelerated cathodic corrosion[41–45]. The Ag-NP catalyst exhibited a minimal change in the

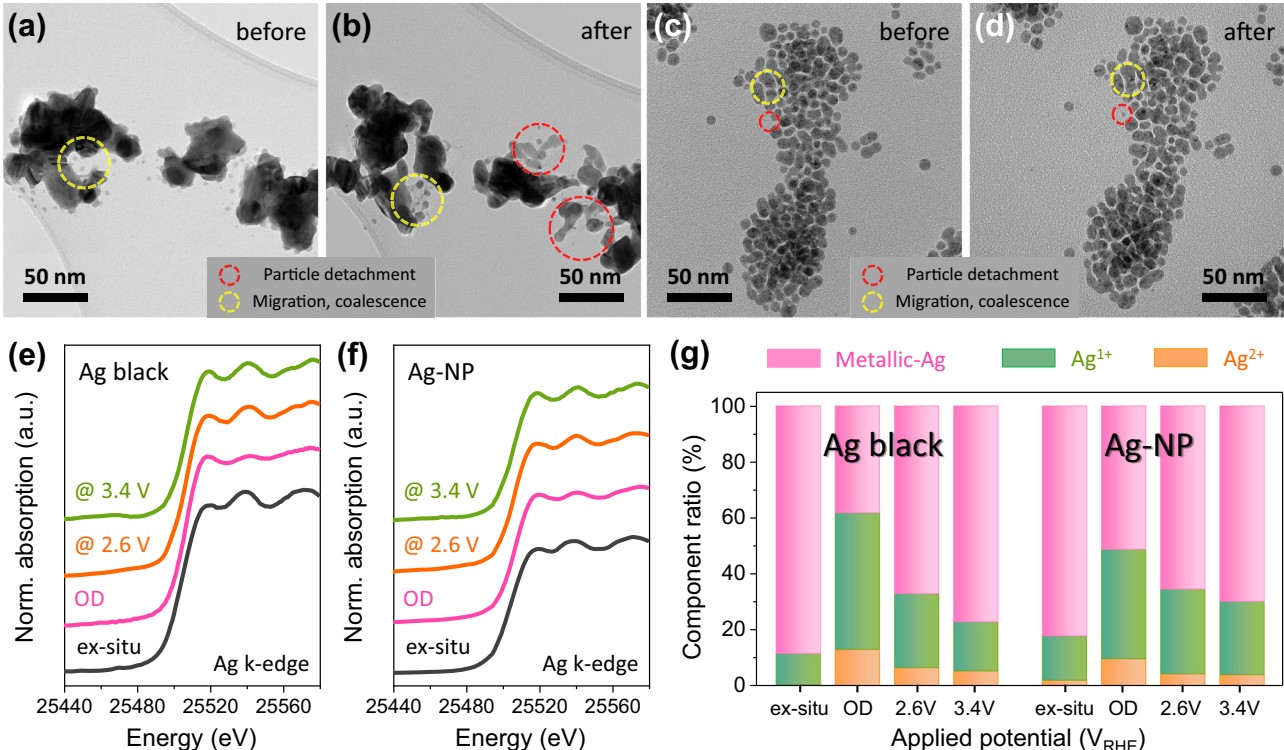

**Fig. 3 | Morphology and phase change during the CO₂RR.** Identical location transmission electron microscopy (IL-TEM) images of the (**a**, **b**) Ag black and (**c**, **d**) Ag-NP catalysts (**a**, **c**) before and (**b**, **d**) after the CO₂RR. **e**, **f** In-situ/*operando* X-ray adsorption near-edge structure (XANES) spectra at the Ag k-edge for the (**e**) Ag black and (**f**) Ag-NP catalysts during the CO₂RR in the MEA-type electrolyzer and its (**g**) oxidation state distribution deconvoluted by linear combination fitting.

oxidation state during the CO₂RR, leading to negligible changes in the morphology due to cathodic corrosion, as confirmed by the IL-TEM results. This was attributed to the inhibition of electrolyte flooding by the lipid ligand, and direct evidence was obtained by observing the electrolyte flooding on the surface of the Ag-based electrodes during the CO₂RR.

The previously identified intrinsic properties (cathodic corrosion and oxidation state) were induced by the extrinsic properties, particularly the electrode-surface hydrophobicity. Hydrophobicity significantly influences the CO₂RR activity; electrode flooding, which generates a hydrophilic surface, hinders the mass transfer of gaseous CO₂[46,47]. Synchrotron X-ray CT was conducted before and after the CO₂RR to compare the electrolyte and CO₂-gas distributions on the cathode surface during the CO₂RR using the Ag black and Ag-NP electrodes. The homemade in-situ/*operando* synchrotron X-ray CT analysis system and zero-gap type CO₂RR device for CT analysis, which was fabricated with a layer of composed parts, are shown in Supplementary Figs. 26, 27, and Fig. 4a. Upon irradiation with a hard X-ray beam, the Ag particles, electrolyte, gaseous CO₂, and carbon layer on the GDL exhibited different transmittances. A comparison of the in-plane cross-sectional tomographs for the Ag black and Ag-NP cathodes indicated the occurrence of electrolyte flooding on the electrode surface at each cathodic overpotential. The segmented electrolyte within the electrode surface is shown in Fig. 4b. Flooding occurred while operating the device at low cathodic overpotentials using the Ag black cathode; flooding increased at high cathodic overpotentials, as indicated by the observation of a significant number of electrolyte droplets. In contrast, when using the Ag-NP cathode, slight flooding was observed at high cathodic overpotentials, with negligible electrolyte droplets at low cathodic overpotentials, similar to the open circuit voltage (OCV) state. Six videos showing a virtual cross-section through the entire image volume of the cathode at

different cell voltages (Supplementary movies 1–6) were recorded; the results verified the nature of flooding on the cathode (local/not local).

WCA analysis was also conducted to investigate the hydrophobicity changes during the CO₂RR using the Ag black and Ag-NP cathodes (Fig. 4c and Supplementary Fig. 28). Prior to the WCA analysis, both cathodes were cleaned with DI water after the CO₂RR to remove excess carbonate ions. As shown in Fig. 4c, the Ag black cathode showed a high WCA (>165°) before the CO₂RR due to the characteristics of its large-size metal particles[48–50]; however, the WCA of the Ag black cathode decreased to 20° after preoxidation. Moreover, it decreased to 80° during the CO₂RR at 3.4 V compared to before the CO₂RR. There were two main reasons for the transformation of the surface from hydrophobic to hydrophilic: a decrease in the particle size due to the splitting and detachment of Ag[42] and the adsorption of the carbonate salt[6,40,51]. A high FE_CO was maintained at low cathodic overpotentials due to the hydrophobicity of the electrode surface; at high cathodic overpotentials, the electrode surface became hydrophilic owing to cathodic corrosion, and carbonate-ion adsorption occurred. A significant decrease in hydrophobicity caused water flooding, hindering the mass transfer of gaseous CO₂ and increasing the HER selectivity. The Ag-NP cathode exhibited a smaller WCA (~130°) than the Ag black cathode, despite the attached lipid ligand (Fig. 4c); however, the decrease in its hydrophobicity during the CO₂RR was negligible in relation to that of the Ag black cathode (WCA ≈ 97°). Owing to the influence of the lipid ligands occupying the corner sites of the Ag-NPs, cathodic corrosion and subsequent carbonate-ion adsorption at high cathodic overpotentials occurred to a lesser extent in the Ag-NP cathode than in the Ag black cathode. In addition, the WCA hardly decreased even with preoxidation treatment (Supplementary Fig. 28). Thus, due to low flooding on the surface of the Ag-NP cathode, the FE_CO was maintained.

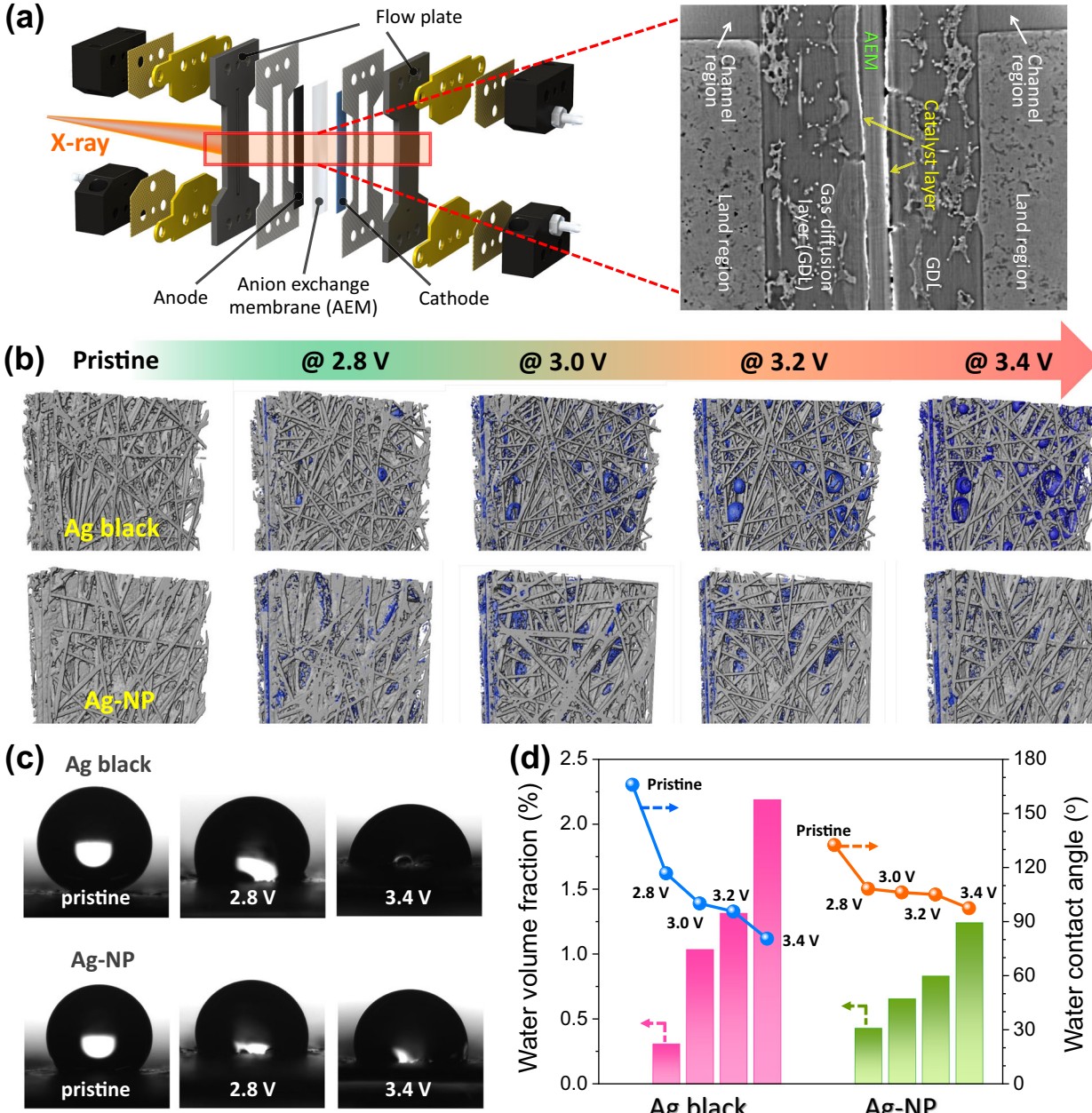

**Fig. 4 | Visual analyses of the influence of electrode hydrophobicity on the CO₂RR. a** A schematic representation of the zero-gap CO₂ electrolyzer used for the in-situ/*operando* synchrotron computed tomography (CT) and cross-section tomography of the Ag-NP cathode. **b** High-magnification synchrotron tomographs of the 3D structure of the Ag black and Ag-NP cathodes at various applied cell

voltages (blue: electrolyte). **c** Water contact angle (WCA) images for the Ag black and Ag-NP cathodes at the initial state, 2.8 V, and 3.4 V. **d** WCA and water volume fraction versus the applied cell voltage in the zero-gap CO₂ electrolyzer for the Ag black and Ag-NP catalysts. The water volume fraction in the electrodes at each applied cell voltage was estimated from the tomographs.

To quantify the relationship between hydrophobicity and flooding, the volume fraction of the electrolyte, according to the WCA at each potential, was considered (Fig. 4d). Considering the current density (i.e., $FE_{CO}$) and long-term stability (Fig. 2a–d), flooding occurred at a WCA of ~100°; the system regularly vented the water and maintained its performance. However, water venting became difficult under hydrophilic conditions (water volume fraction > 1.5%), causing low performance. Therefore, to maintain a high partial current density of CO, a hydrophobic catalyst maintaining its WCA above 100° was required.

Thus, particle splitting/detachment occurred during the CO₂RR over the intrinsic hydrophobic Ag black catalyst, as indicated by the spectroscopy and ex-situ analysis results. Consequently, the electrode surface became hydrophilic (as indicated by real-time observations),

and the subsequent carbonate-ion adsorption increased its hydrophilicity, intensifying flooding. Particle splitting/detachment was significantly suppressed in the presence of the lipid ligand. Therefore, although the initial hydrophobicity of the Ag-NPs was low, its hydrophobicity was maintained even at high cathodic overpotentials.

## Discussion

Lipid ligands conferred extrinsic hydrophobic properties to the Ag-NPs synthesized in this study, enabling the maintenance of the $FE_{CO}$ of the Ag-NP cathode even at high cathodic overpotentials. The Ag black catalyst exhibited high intrinsic hydrophobicity; however, its particle size decreased because of the splitting/detachment of the Ag particle during the CO₂RR, transforming the electrode surface from hydrophobic to hydrophilic. Subsequently, the hydrophilicity of the surface

was enhanced by carbonate-ion adsorption. Thus, the mass transfer of gaseous $CO_2$ was hindered owing to electrolyte flooding, thereby reducing the partial current density of CO. In contrast, due to the extrinsic hydrophobic properties of the Ag-NPs (derived from the lipid ligands), the particle splitting/detachment phenomenon was significantly suppressed on the Ag-NP cathode during the $CO_2RR$, and its hydrophobicity was maintained. The mass transfer of gaseous $CO_2$ was maintained in the catalytic range of hundreds of nanometers, thereby smoothly forming a triple-phase boundary. Therefore, the partial current density of CO was maintained even at high cathodic overpotentials. A similar phenomenon occurred in Ag-PTFE electrodes, but a severe current fluctuation occurred due to the random distribution of PTFE. Thus, in the zero-gap $CO_2$ electrolyzer, the Ag-NP cathode with an Ag loading amount of $0.3\,mg\,cm^{-2}$ exhibited a high $FE_{CO}$ (>90%), with maximum CO partial current density of $298.39\,mA\,cm^{-2}$ at a cell voltage of 3.4 V. This study describes a strategy to incorporate intrinsic and extrinsic (particularly hydrophobicity retention) properties in an advanced catalyst structure to facilitate a high partial current density and durability.

## Methods

### Materials

Silver nitrate ($AgNO_3$, Sigma-Aldrich, 99.0%), oleylamine (OLA, Sigma-Aldrich, 70%), oleic acid (Sigma–Aldrich, 90%), tetramethylammonium hydroxide pentahydrate (TMAH, Sigma-Aldrich, 97%), acetone (DAE-JUNG, 99.5%), n-hexane (SAMCHUN, 95.0%), and isopropyl alcohol (DAEJUNG, 99.5%) were used as purchased without further purification. A commercial Ag black catalyst was purchased from Alfa Aesar.

### Synthesis

TMAH-treated Ag-NPs were synthesized according to a previously reported method, with some modifications[52,53]. First, a 100-mL three-neck flask was loaded with 2 mmol of $AgNO_3$ and 40 mL of OLA. After degassing for 15 min, the solution was heated to 60 °C under $N_2$ gas to dissolve the silver salt. Subsequently, the solution was heated to 160 °C at a rate of 10 °C/min to synthesize Ag-NPs with sizes of 7.36 nm. After heating, the required temperature was maintained for 1 h to allow sufficient growth of the Ag-NPs. Subsequently, the heating mantle was removed, and the solution was cooled in a water bath. For purification, 80 mL of acetone was added to the crude solution, which was centrifuged at $2010 \times g$ for 3 min. After centrifugation, the supernatant was discarded, and the precipitate was dispersed in n-hexane. This purification process was repeated at least twice. For TMAH treatment, the precipitated Ag-NPs were mixed with 0.5 wt.% TMAH in DI water and sonicated for 2 h, followed by centrifugation and redispersion in DI water twice. Subsequently, the Ag-NPs were dried under vacuum for 10 min and dispersed in n-hexane/isopropyl alcohol (1:1, v/v) for electrode fabrication.

### Preparation of Ag black, Ag-PTFE, and Ag-NP electrodes

Ag black, Ag-PTFE, and Ag-NP electrodes were fabricated by spraying the catalyst ink onto the GDL (Sigracet 39 BB from the Fuel Cell Store). The catalyst ink for the Ag black catalyst was ultrasonically blended with isopropyl alcohol, 5 wt.% Nafion solution (Sigma–Aldrich), and commercial silver nanopowder (Alfa Aesar), while the Ag-NP catalyst ink was sprayed onto the GDL after adding 5 wt.% Nafion solution. The catalyst ink for Ag-PTFE was manufactured by the same method used for Ag black, except a PTFE solution was blended instead of a Nafion solution.

### Electrochemical $CO_2RR$ single-cell tests

The detailed scheme of an AEM zero-gap-type $CO_2$ electrolyzer is shown in Supplementary Figs. 11 and 12. Membrane electrode assemblies (MEAs) were fabricated using the catalyst-coated electrode method. The catalysts were sprayed onto the GDL using spray-gun and the

geometric electrode area of each MEA was 10 $cm^2$. Commercial $IrO_2$-sprayed $Pt/TiO_2$ mesh electrodes (Alfa Aesar, loading = $1.0\,mg\,cm^{-2}$) were used as anodes in all single-cell tests. Before assembly, the AEM (dioxide materials, X37-50 Grade RT) was pretreated in a 1 M KOH solution for 48 h and washed several times with deionized water. To prepare the oxide-derived Ag-based electrodes, a reverse voltage of 3 V was applied to the zero-gap electrolyzer for 2 min. The preoxidation process was performed to remove impurities remaining on the surface of the Ag-NPs, and there was no change in oleylamine on the Ag-NP surface after the preoxidation process (Supplementary Fig. 10). Subsequently, a 0.1 M $KHCO_3$ solution was supplied to the anode side, while 200 sccm of humidified $CO_2$ gas at 50 °C was inserted into the cathode side. All electrochemical experiments were performed using a VSP potentiostat (Bio-Logic) with a 10 A booster. The $CO_2RR$ was performed for 18 min at each cell voltage. The mean value was used for the performance of CO2RR electrolyzer by cell potential for all samples, and the results are summarized in Supplementary Table 2. The iR-compensation was not performed in all electrochemical experiments.

Gas chromatography (GC, Agilent 7890 A) was conducted for output-gas analysis. The GC system was connected to a cathode water trap, which was attached to the cathode outline. Ultrahigh-purity He gas was used as the carrier gas; a flame ionization detector (FID) with a methanizer and a thermal conductivity detector (TCD) were used in the GC system. The FID and TCD detectors were used to detect hydrocarbon products and hydrogen ($H_2$) gas, respectively; the methanizer was used to enhance CO detection by the FID detector. Considering the diffusion time of the generated gases in the device for GC, the measurements were started 9 min after the reaction started for each voltage. The measurement time was 13.5 min.

The Faradaic efficiencies of $H_2$ and CO gases were calculated using the following equation:

$$FE_{product}(\%) = \frac{i_{product}}{i_{total}} \times 100 = \frac{V_{product} \times Q \times \frac{2Fp}{RT}}{i_{total}} \times 100 \qquad (1)$$

where $Q$ is the flow rate of the total product, $F$ is the Faradaic constant ($96485\,C\,mol^{-1}$), $p$ is the pressure, $T$ is room temperature (298 K), and $R$ is the ideal gas constant ($8.314\,J\,mol\,K^{-1}$). The partial currents of the products were calculated from the volume of the product obtained from the GC peak of each product. The experimental results of the AEM zero-gap-type $CO_2$ electrolyzer are summarized in Supplementary Table 2.

### Physical characterization

The size distribution and microstructure of the Ag-NPs were analyzed through high-resolution transmission electron microscopy (HR-TEM, Titan at 300 kV, FEI Co., USA). Ag-NP inks diluted to 1/10 concentration using n-Hexane were drop-cast on an ultrathin carbon grid under a vacuum filtration to prevent stacking of particles. After measuring the long-term experiment, the catalyst drop-casted on the GDL was dispersed in isopropyl alcohol via sonication. TEM analysis was performed by drop casting the collected catalysts onto a TEM grid. Wide-angle XRD (Bruker D8 Advance instrument, Cu Kα radiation) was employed to determine the crystal structure of the Ag-NP catalyst. After the AEM zero-gap-type $CO_2$ electrolyzer test, the Ag catalyst-coated GDL was measured to observe the change in crystallinity after the $CO_2RR$. XRD tests were conducted at a $2\theta$ angle of 30 to 80° with a scan rate of 2° $min^{-1}$. A drop-shape analyzer (Kruss DSA 100) was used to measure the contact angle of deionized water at each $CO_2RR$ potential. WCA was measured after the AEM zero-gap-type $CO_2$ electrolyzer test for each cell voltage. The cathode electrode for wide-angle XRD and WCA analysis was fabricated using the same procedure used for the single-cell test. Partial ligand exchange from oleylamine to TMAH was observed with an FTIR spectrometer (Frontier, Perkin-Elmer Co., USA). Ag-NPs before and after TMAH

treatment were spin-cast onto a double-side polished Si wafer for FTIR measurements. Preoxidation and the $CO_2RR$ for ex-situ FT-IR measurements were performed using a VSP potentiostat (Bio-Logic) in a conventional three-electrode system with a homemade poly-ether ether ketone (PEEK) cell equipped with Ag/AgCl (3 M NaCl) and a graphite rod as the reference and counter electrodes, respectively. The presence of oleylamine was confirmed by $^1H$ NMR in $CDCl_3$ using a solution-state NMR 600 MHz spectrometer (Agilent) (Supplementary Fig. 8).

### IL-TEM analysis

All electrochemical experiments for IL-TEM analysis were performed in a conventional three-electrode system with a homemade PEEK cell. Ag/AgCl (3 M NaCl) and a graphite rod were used as the reference and counter electrodes, respectively. Ag black and Ag-NP inks were drop-cast on a holey carbon-coated Au grid (Agar Scientific, H7 finder grids) and placed on a rotating disc electrode (RDE; VSP, Bio-Logic Science Inc.) in a customized holder, and the grid was fixed by screwing a PEEK cap to secure electrical contact. The $CO_2RR$ performance of the bare Au grid is presented in Supplementary Fig. 29, which was negligible compared to the $CO_2RR$ performance of the Ag-NP and Ag black-coated Au grids. The electrochemical measurements were carried out in $CO_2$-saturated 0.1 M $KHCO_3$, and the potential was converted to the RHE scale by applying the following equation:

$$E_{RHE} = E_{Ag/AgCl} + 0.209\,V + 0.059\,V \times pH \qquad (2)$$

TEM images were collected at the same location of the Au grid, and the morphology of the catalysts was compared before and after the $CO_2RR$ by relying on the applied potentials ($-1.0\,V_{RHE}$) and the reaction time (4 h) at a rotation speed of 1600 rpm.

### APT characterization

APT specimens were prepared by FIB milling (Helios Nanolab 450, FEI). The Ag-NPs were deposited within the Ni matrix according to the following procedure comprising three steps. First, the Ag-NPs were spin-coated on a 60-nm-thick Cr-deposited Si substrate. Subsequently, a 20-nm-thick layer of Cr was deposited at a rate of $0.1\,A\,s^{-1}$. Finally, a low clean-up voltage (5 kV) was applied to the sharpened APT specimens to restrict the implantation of $Ga^+$ ions during FIB milling to a minimum level. A CAMECA LEAP™ 4000X HR system was used in pulsed laser mode (at a detection rate of 0.3%, a base temperature of 65 K, a laser pulse energy of 50–60 pJ, and a pulse frequency of 125 kHz) for APT. Data reconstruction and analyses were performed using the commercial Imago Visualization and Analysis System (IVAS) 3.8.2 software developed by CAMECA Instruments. All three-dimensional atom maps presented in this study were reconstructed using a standard voltage reconstruction protocol[54].

### In-situ/operando X-ray analyses

Ag k-edge hard-X-ray absorption spectroscopy (XAS) of the Ag catalysts was conducted at the 10 C beamline of the Pohang Acceleration Laboratory (PAL) with a homemade electrochemical zero-gap single cell. For in-situ/*operando* hard XAS, a $1$-$cm^2$ hole was drilled in the bipolar plates and covered with a Kapton film (to allow transmission of the X-ray beam). The operating conditions were the same as those used in the single-cell experiment; hard-XAS was performed in fluorescence-collection mode using a Si (311) monochromator. The photon flux of the incoming hard X-ray was approximately $1 \times 10^{11}$ photons per second, and the nominal beam size was 1 mm × 1 mm. The reference spectra of the Ag foil, Ag(I) oxide and Ag(II) oxide were obtained in transmission mode in an Ar-filled chamber under ambient pressure at room temperature (Supplementary Fig. 25: XANES spectra of Ag foil, Ag(I) oxide and Ag(II) oxide). The measured spectra were calibrated using Ag foil to ensure zero shift in the k-edge energy, and

the XANES data were fitted by the LCF method using Athena software (Demeter ver. 0.9.20). A description of the LCF method is provided in Supplementary Note 1, and the fitting results are tabulated in Supplementary Table 3.

Micro-scale in-situ/*operando* CT was performed at the 6 C beamline of PAL; the in-situ/*operando* CT setup is shown in Supplementary Figs. 26 and 27. The cathode electrode for CT analysis was fabricated using the same procedure used for the single-cell test. The geometric electrode area of each GDL electrode was 1 cm × 3 cm. A polychromatic X-ray beam centered at 34 keV was used to achieve a high signal-to-noise ratio. The stage required 2 s to accelerate to the required rotation velocity; 1000 radiographs were recorded over 180° rotations within 40 s of attaining a constant angular velocity. For in-situ/*operando* CT analysis, each potential was applied for 10 min. Octopus 8.7 software was used to reconstruct the in-situ/*operando* CT scans, while Avizo 3D (Thermo Fisher Scientific, Hillsboro, OR, FEI Visualization Sciences Group) was used to analyze the reconstructed images in 3D.

## Data availability

The data generated in this study are provided in the Source Data file. Source data are provided with this paper.

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

## Acknowledgements

This work was supported by institutional program grants from the Korea Institute of Science and Technology and "Carbon to X Project" (Project No. 2020M3H7A1098229) through the National Research Foundation (NRF) funded by the Ministry of Science and ICT, Republic of Korea. This research was also supported by a National Research Council of Science & Technology (NST) grant by the Korean government (MSIT) (No. CAP21011-100) and a National Research Foundation of Korea (NRF) grant funded by the Korean government (MSIT) (NRF-2021R1A2C2093467). We are grateful for

access to the 10 C and 6C PAL beamlines to conduct hard XAS and CT, respectively. We are thankful to Dr. In-Kyoung Ahn and Dr. Taekeun Kim from LG Chem's Platform Technology Research Center for their assistance with the $CO_2$ electrolysis measurements.

## Author contributions

H.-S.O., W.H.L., and G.W.H conceived the idea, designed the experiments, and supervised the work. Y.-J.K. and J.J. synthesized the catalysts, analyzed the data, and wrote the manuscript. C.L. conducted the electrochemical and in-situ/*operando* experiments and wrote the manuscript. M.G.K., B.K.M., J.-Y.C., and J.Y.L performed the XANES and APT analyses. T.-Y.S. and K.-Y.L. contributed to the catalyst synthesis. T.N contributed to the $CO_2$ electrolysis measurements. All authors reviewed the manuscript and agreed with its content.

## Competing interests

The authors declare no competing interests.
