## [Peer Review File · Nature Communications]

REVIEWER COMMENTS

Reviewer #1 (Remarks to the Author):

The current work by Hyung-Suk Oh and co-workers tackles the cathode flooding challenge observed during the electrochemical CO₂ reduction using a zero-gap electrolyzer system by designing partially ligand-derived Ag nanoparticles (Ag-NPs) with extrinsic hydrophobic properties.

The performance of the Ag-NPs electrodes was compared to the one observed with Ag black electrodes. It was demonstrated that even though the Ag black electrodes exhibited high intrinsic hydrophobicity and presented higher partial current densities for CO, they were less stable due to strong cathodic corrosion, which promotes flooding during CO₂ electroreduction.

The superior performance of the Ag-NPs catalyst was attributed to the presence of hydrophobic ligands on their surfaces, which improve the reaction performance in terms of high faradaic efficiency for CO, high partial current density, long-term stability, and less cathodic corrosion. More importantly, it was demonstrated through operando synchrotron tomography that the presence of hydrophobic ligands on the Ag-NPs avoids cathode flooding at high-applied overpotentials.

Most of the approaches to control the flooding/electrolyte management of gas diffusion electrodes involve using hydrophobic materials such as PTFE. However, this work proposed a new method that consists of designing silver nanomaterials with hydrophobic properties to improve the electrolyte management of cathodes for CO₂ electroreduction.

The study is thoroughly conducted and relevant to the wider CO₂ reduction community as the field moves toward an increased focus on the study of nanostructured catalysts. Taking into account the novelty and importance of these results, I can recommend the manuscript for publication after the following points are addressed:

1. It was shown that the native ligands on the Ag-NPs are removed by TMAH treatment. Can the authors estimate how the electrochemically active surface area of the Ag-NPs is affected by this treatment?
2. The authors mentioned that the size and shape of the Ag-NPs are not significantly affected by the reduction of the native ligands. Can the characterization of the pristine Ag-NPs be included in the supplementary information to confirm the previous statement?

3. It is mentioned in the manuscript that the ligands are attached to the edge of the nanocrystal, and in another paragraph is mentioned that the ligands occupy the defect sites of the Ag-NPs. Can the authors mention which sentence is correct or if both sentences are correct? In addition, is there experimental proof or references to confirm those statements?
4. The current catalyst is an oxide-derived Ag-NPs catalyst. However, no characterization of the Ag-NPs electrodes after the oxidation step is provided. How is the hydrophobicity of the Ag-NPs electrode after the oxidation step? Do the WCA experiments using Ag-NPs electrodes further confirm that the lipid ligand is maintained?
5. Mention the duration of the electrolysis experiments performed to plot the Fig. 2a. How frequently were the gaseous products analyzed? Moreover, no error bars are observed in that plot.
6. Mention the duration or the applied charge density of the electrolysis experiments performed to obtain the IL-TEM images (Fig. 3a-d). Furthermore, there is no experimental description of the IL-TEM experiments. What kind of cell was used to carry out those experiments?
7. The colors of the Metallic-Ag, Ag⁺, and Ag²⁺ mentioned in the caption of Fig. 3g (LFC) do not correspond to the colors observed in the plot.
8. What was the electrolysis time or applied charge (at the different applied potentials) of the operando synchrotron computed tomographs observed in Fig. 4b?
9. On page 12, it is mentioned that "The segmented electrolyte within the electrode surface is shown in Fig. 4d", I think it is 4b instead of 4d.
10. Can the authors comment on the differences between local/not local flooding of the cathode?
11. On page 15, it is mentioned that "Electrochemical CO₂RR single-cell tests. The detailed scheme of an AEM zero-gap-type CO₂ electrolyzer is shown in Supplementary Fig. 3 and Supplementary Fig. 4." It should be Supplementary Fig. 5 and Supplementary Fig. 6.
12. In the experimental description, it is not clear what "200 cm of humidified CO₂ gas" means. Is it SCCM instead of cm?
13. On page 17, it is mentioned that "the in-situ/operando CT setup is shown in Supplementary Fig. 6 and Supplementary Fig. 7." It should be Fig. 4a, supplementary figure 14, and supplementary figure 15.
14. The caption of Supplementary Figure 7 is incorrect. (b) CO faradaic efficiency instead of CO partial current density and (c) CO partial current density instead of H₂ partial current density.

Reviewer #2 (Remarks to the Author):

The present work of Hyung-Suk Oh and co-workers reports the CO₂ electrolysis performance of Ag nanoparticles obtained by a colloidal synthesis followed by partial ligand removal. The importance was given to the effect of the synthesis-derived-ligands on the long term stability for CO₂ in effort to minimize electro-wetting at high current density testing. Most of the figures are clear and there are several operando methods (Synchrotron X-ray CT and XANES) to integrate visual aid for the catalytic activity and stability which are novel to the field of CO₂ electrolysis. Yet, the info provided in the methods and SI are not detailed enough and needs further inputs. There are some logical leaps in the interpretation of the data, especially amongst the samples. At the current state, the conclusions do not seem to present a major advancement in the current state-of-art. Thus, the work is recommended to be go under significant revision after considering the following questions and comments;

Questions

1. The sentence in page 3 claims that

a. "... electrocatalysts with hydrophobic properties have not yet been reported..."

Several groups have demonstrated the influence of long-chain ligands derived from the colloidal nanoparticle synthesis of Cu, Ag and CuAg such as oleylamine¹ and nitro-containing ligands², displaying a hydrophobic property in CO₂ electrolysis by capacitance and zeta-potential measurements, respectively. Therefore the authors need to rephrase this sentence by assessing the results in the established literature.

b. "... Furthermore, the lipid ligand inhibits Ostwald ripening and sintering of the electrocatalyst during the CO₂RR, thereby facilitating the maintenance of the fine nanoparticle size..."

This phenomenon (catalyst sintering) can be explained by the collapse of the ligand upon its introduction into an aqueous solution. In particular, CO₂ electrolysis is known to be highly alkaline at the local reaction environment, pH > 11 at high current densities > 200 mA/cm² ³. In such conditions, the ligands may undergo hydrolysis and the conformal arrangement of Ag-NPs would collapse. This has been demonstrated by ex-situ ligand exchange studies in this article: ACS Catal. 2020, 10, 22, 13468–13478 ¹

2. Colloidal synthesis in OLA-type solutions require hazardous chemicals during the synthesis and the purification steps. On top of that, the yield of the synthesis are usually 30 % and below in terms of grams of metal used. After 50 hours, the selectivity and partial CO current density are not drastically different between the Ag-NPs and commercial Ag-PTFE samples. Could the authors provide further discussion on the novelty in that aspect?

3. Referring the line in Page 5: “ ... To analyze the hydrophobic properties while maintaining the morphology of the catalyst, by leaving a part of the ligand, physical property analysis was performed using Ag-NP treated with TMAH for 2 h. ...”

a. Figure S2 shows the HR-TEM images of the “OLA-capped Ag-NPs” after 5 h of TMAH treatment to remove OLA, which results with the collapse of the conformal arrangement and the size of Ag-NPs. Hence, an alternative of 2 h TMAH treatment was suggested to preserve the NP stability. However, the evidence for the stability after 2h TMAH treatment is limited – so far, this evidence is limited to three individual Ag particles in Figure S3. Could the authors please provide further evidence with HR-TEM images of the selected process (2h-TMAH), comparable to the images in Figure S2 (or in Fig.1a)? which are the TEM images used to build the histograms of 2h-TMAH sample?

4. Eventhough Atom Probe Tomography (APT) technique seems to be a strong indicator of the presence of N-containing ligands (such as oleylamine-OLA and tetramethylammonium-TMAH), a conclusive analysis for the presence of the surface groups would be the Fourier Transform Infrared Spectroscopy (FTIR) or the analysis of the colloidal solution by Nuclear Magnetic Resonance (NMR). The N-signal observed by APT could be derived from OLA and/or TMAH, therefore could the authors reach a conclusion at this end and provide stronger evidence about the content of the surface chemistry?

5. There is a logical leap of using ligands, and the going to partial ligand-exchange/removal and then an oxidation step... What is the motivation for the authors to form an oxide-derived Ag-NPs, instead of the as-synthesized Ag-NPs?

6. FTIR spectra of the OD-Ag NPs were given Figure S4. The preparation method of this sample for the FTIR analysis is not clear, please specify in the experimental section. The sustanion chemistry may also give signals around 2600 – 3200 cm^{-1} wavenumbers⁴. After the disassembling the MEA, could the authors be certain that those peaks are only belong to the synthesis ligands of the Ag-NPs? Is there a control sample where commercial Ag-NPs were analyzed by FTIR after an oxidation step in a similar MEA assembly?

7. Regarding Figure S8,

a. The Faradaic efficiencies and partial current densities do not match. For ex. at 3.6 V, Ag black shows 70 % FE of H₂ and 500 mA/cm^2 total cell current, which should be equal to 350 mA/cm^2 partial current for H₂. Yet, Figure s8c shows only 275 mA/cm^2 . Could you please explain the inconsistency in the efficiency data? Please present a table of results for the tests conducted.

b. There are not any error bars in any of the data. Is this a single test run? Is there any deviation amongst the GC injections – whether there are multiple injections or not? And are there any duplicate or triplicate tests to minimize the deviation?

8. In page 7, please clarify or rephrase this statement, as it is inconclusive with the prior sentence. “...Therefore, the partial current density of CO exhibited by the Ag-NP catalyst at low cathodic overpotentials was significantly smaller than the values at high cathodic overpotentials...”

9. The size of the Ag-black particles need to be clarified, as the authors claim a higher activity for Ag-NPs due to its “moderate size between 5 – 10 nm”. How is this an indication of “...extrinsic effect of the lipid ligand..”, please clarify?

10. In Figure S9,

a. The partial current densities do not add up to the total value. For ex. at 3.4 V, the total j is around 300 mA/cm², while the j_{CO} is between 220 – 250 mA/cm² and j_{H_2} is almost zero, please clarify the breakdown of the calculation, preferably with table of results?

11. Regarding line “ .. This is due to a decrease in the active surface area at the expense of suppressing the flooding phenomenon at a high cell voltage...”

a. Are there any ECSA measurement to support this claim?

b. For Ag-PTFE, the total cell current is less compared to Ag-NPs, which points to a higher resistance under fixed voltage testing. Hence, the authors need to consider the electrical resistance contribution of the PTFE additive instead.

12. In Figure 2,

a. In Fig 2b, the cell voltage shows a steady increase 80th hour from 3.1 V to ca. 3.5 V, and then experiences a sudden drop back to 3 V. Could the authors please explain why such a change has occurred?

b. In Figure 2c-d & Fig S10, did the authors facilitated water removal externally by flushing with a liquid or inert gas media in any of the tests, at any time point? If yes, please specify the details in the experimental section.

c. In comparison of the j_{CO} performance of Ag-PTFE vs. OD-Ag-NPs treated with TMAH, the starting and ending partial CO current densities are identical (350 and 225 mA/cm²) along the 50 hours stability test. Yet, the water management property of the Ag-PTFE vs. OD-Ag-NPs treated with TMAH are different regarding the observed current spikes (150 vs 200 mA/cm² min. point). The authors need to acknowledge the similarity in the performance (y-axis is over-stretched in the Fig S10 in comparison to Fig. 2c!) but they need to provide further evidence on, what sort of mechanism derives the difference in the local water management? (the last sentence of the paragraph in page 8 needs further evidence / discussion)

13. About the IL-TEM,

a. The details of the sample preparation for the IL-TEM is not mentioned at any part of the manuscript or SI. Please clarify how are those steps following the disassembly of the MEA and preparing a TEM specimen for the post-mortem analysis?

b. Please provide information about the cell configuration and electrodes used for the ex-situ analysis here.

14. In page 10, regarding line “... They expose the high-crystallinity (111) plane, inducing excellent catalytic activity toward the CO₂RR under cation-assisted conditions^{32,33}, enhancing the CO₂RR

activity without cathodic corrosion....” This postulation needs further evidence such as; the authors need to justify this postulation with a before and after sample analysis with XRD pattern. The Ag (110) facet was shown by both experimental and theoretical work to be more active than the very stable (111) facet⁵.

15. The part about the XANES spectra in page 10, 2nd paragraph needs further discussion. It is advised to provide a brief information about the Operando X-ray absorption spectroscopy for the broader audience, as in Ref 5. The parameters such as the sample tilt needs to be addressed as it effects the penetration depth of the X-rays from 2 nm to 12 um at 0° to 45° angles.

a. The analysis of the results are not easy to follow such as the meaning of LCF is not explained. For ex. the quantification methods and the data analysis details needs to be reported in the SI and/ or the fitting results of the reference metal, metal-oxides and samples of Fig 3 e-f need to be reported in a table.

(see example <https://www.rsc.org/suppdata/c8/ta/c8ta10412c/c8ta10412c1.pdf>)

b. Could the authors elaborate on whether the beam alone, at the intensity used & timescale of the reaction, was capable of reducing the silver or not?

16. WCA of 3.4 V is shown in Fig 4c but its value is missing in the chart of Fig 4d, instead 3.8 V was shown. Is this intentional or typo error? What is the contact angle of the electrode after 3.4 V?

References

1. Irtem, E.; Arenas Esteban, D.; Duarte, M.; Choukroun, D.; Lee, S.; Ibáñez, M.; Bals, S.; Breugelmans, T., Ligand-Mode Directed Selectivity in Cu–Ag Core–Shell Based Gas Diffusion Electrodes for CO₂ Electroreduction. *ACS Catal.* 2020, 10 (22), 13468-13478.

2. Pankhurst, J. R.; Guntern, Y. T.; Mensi, M.; Buonsanti, R., Molecular tunability of surface-functionalized metal nanocrystals for selective electrochemical CO₂ reduction. *Chem. Sci.* 2019, 10 (44), 10356-10365.

3. Burdyny, T.; Smith, W. A., CO₂ reduction on gas-diffusion electrodes and why catalytic performance must be assessed at commercially-relevant conditions. *Energy Environ. Sci.* 2019, 12 (5), 1442-1453.

4. Kaczur, J. J. A. U. Y. H. L. Z. S. S. D. A. U. M. R. I. T. I. A. R. o. t. U. o. I. C [Online], 2020.

5. Firet, N. J.; Blommaert, M. A.; Burdyny, T.; Venugopal, A.; Bohra, D.; Longo, A.; Smith, W. A., Operando EXAFS study reveals presence of oxygen in oxide-derived silver catalysts for electrochemical CO₂ reduction. *J. Mater. Chem. A* 2019, 7 (6), 2597-2607.

Authors' response to the reviewer's comments

Reviewer: 1

Recommendation: The current work by Hyung-Suk Oh and co-workers tackles the cathode flooding challenge observed during the electrochemical CO₂ reduction using a zero-gap electrolyzer system by designing partially ligand-derived Ag nanoparticles (Ag-NPs) with extrinsic hydrophobic properties.

The performance of the Ag-NPs electrodes was compared to the one observed with Ag black electrodes. It was demonstrated that even though the Ag black electrodes exhibited high intrinsic hydrophobicity and presented higher partial current densities for CO, they were less stable due to strong cathodic corrosion, which promotes flooding during CO₂ electroreduction.

The superior performance of the Ag-NPs catalyst was attributed to the presence of hydrophobic ligands on their surfaces, which improve the reaction performance in terms of high faradaic efficiency for CO, high partial current density, long-term stability, and less cathodic corrosion. More importantly, it was demonstrated through operando synchrotron tomography that the presence of hydrophobic ligands on the Ag-NPs avoids cathode flooding at high-applied overpotentials.

Most of the approaches to control the flooding/electrolyte management of gas diffusion electrodes involve using hydrophobic materials such as PTFE. However, this work proposed a new method that consists of designing silver nanomaterials with hydrophobic properties to improve the electrolyte management of cathodes for CO₂ electroreduction.

The study is thoroughly conducted and relevant to the wider CO₂ reduction community as the field moves toward an increased focus on the study of nanostructured catalysts. Taking into account the novelty and importance of these results, I can recommend the manuscript for publication after the following points are addressed:

Comment 1: It was shown that the native ligands on the Ag-NPs are removed by TMAH treatment. Can the authors estimate how the electrochemically active surface area of the Ag-NPs is affected by this treatment?

Response: We appreciate reviewer's valuable comment. We measured cyclic voltammetry (CV) of the Ag-NP catalysts before and after TMAH treatment and compared the surface area of Ag through the oxidation peak of Ag. According to the Randle-Sevcik equation, the peak current is proportional to the active surface area, so it can be seen that the active surface area increased 3 times after TMAH treatment. We added CV data and related sentences to the revised manuscript.

Supplementary Figure 2. Cyclic voltammograms of Ag-NP catalysts with and without 2 h of TMAH treatment in 0.1 M KHCO₃ solution obtained at a scan rate of 50 mV s⁻¹.

Main manuscript Page 5, Line 4: Change in electrochemical active surface area (ECSA) with and without TMAH treatment was analyzed for Ag/Ag⁺ oxidation reaction (Supplementary Fig. 2). According to the Randle-Sevcik equation, the peak current is proportional to the ECSA. The ECSA of Ag-NPs increased 3 times due to TMAH treatment for 2 h.

Comment 2: The authors mentioned that the size and shape of the Ag-NPs are not significantly affected by the reduction of the native ligands. Can the characterization of the pristine Ag-NPs be included in the supplementary information to confirm the previous statement?

Response: We are thankful to the reviewer for this valuable comment. We added HR-TEM images of pristine Ag-NPs in the revised manuscript.

Supplementary Figure 6. HR-TEM image of pristine Ag-NP catalyst

Main manuscript Page 5, Line 15: The HR-TEM images indicated that the size and shape of the Ag-NPs were not significantly affected by a reduction of the native ligands (Supplementary Fig. 6: HR-TEM image of pristine Ag-NPs).

Comment 3: It is mentioned in the manuscript that the ligands are attached to the edge of the nanocrystal, and in another paragraph is mentioned that the ligands occupy the defect sites of the Ag-NPs. Can the authors mention which sentence is correct or if both sentences are correct? In addition, is there experimental proof or references to confirm those statements?

Response: We appreciate reviewer's valuable comment. Defect site was an erroneous expression, so the word has modified to edge site in the revised manuscript.

Main manuscript Page 14, Line 5: Owing to the influence of the lipid ligands occupying the edge sites of the Ag-NPs, cathodic corrosion and subsequent carbonate-ion adsorption at high cathodic overpotentials occurred to a comparatively lesser extent in the Ag-NP cathode than in the Ag black cathode.

Comment 4: The current catalyst is an oxide-derived Ag-NPs catalyst. However, no characterization of the Ag-NPs electrodes after the oxidation step is provided. How is the hydrophobicity of the Ag-NPs electrode after the oxidation step? Do the WCA experiments using Ag-NPs electrodes further confirm that the lipid ligand is maintained?

Response: Thank you for bringing this issue to our attention. Ag-NPs showed little change in contact angle after oxidation treatment (from 132.45° to 125.37°), but Ag black was greatly reduced (from 165.90° to 145.37°). It is possible to indirectly confirm that the lipid ligand was maintained even after pre-oxidation treatment. The WCA images after oxidation treatment and related sentences were added to the revised manuscript.

Supplementary Figure 27. Water contact angle (WCA) analysis of (a) Ag black and (b) Ag-NP cathodes after pre-oxidation and CO₂RR at 3.0 V and 3.2 V, respectively.

Main manuscript Page 13, Line 25: however, the WCA of the Ag black cathode decreased to 20° after pre-oxidation. Moreover, it decreased to 80° during CO₂RR at 3.4 V compared to before CO₂RR.

Main manuscript Page 14, Line 8: In addition, WCA hardly decreased even with pre-oxidation treatment (Supplementary Fig. 27).

Comment 5: Mention the duration of the electrolysis experiments performed to plot the Fig. 2a. How frequently were the gaseous products analyzed? Moreover, no error bars are observed in that plot.

Response: We are thankful to the reviewer for this valuable comment. We added sentence about the duration of electrolysis and GC measurement conditions to the experimental methods section. In addition, we expressed the CO₂RR performance of the catalysts as average values for three experiments, and added error bars to the corresponding figures.

Fig. 2. Single-cell performance of the Ag-NP catalyst. (a) Selectivity of CO, and (b) CO partial current density versus applied cell voltage in a zero-gap CO₂ electrolyzer for the Ag black and Ag-NP catalysts. (b, c) Durability test results of the Ag-NP catalyst in the zero-gap CO₂ electrolyzer at (b) 150 mA cm⁻² for 100 h and (c) 3.4 V for 50 h. (d) Durability test result of the Ag black catalyst in the zero-gap CO₂ electrolyzer at 3.4 V for 15 h. Selectivity of CO and H₂ measured during the durability tests. Electrodes of the CO₂ electrolyzer were prepared with 0.3 mg cm⁻² of Ag catalysts on a 10-cm² gas diffusion layer (GDL) on the cathode side. (e) Schematics (low and high magnifications) of the triple phase boundary for the hydrophilic Ag black and hydrophobic Ag-NP catalysts in the CO₂ electrolyzer.

Main manuscript Page 16, Line 23: CO₂RR was performed for 18 min at each cell voltage.

Main manuscript Page 16, Line 30: Considering the diffusion time of generated gases in the device to GC, the measurements was started 9 min after the reaction start for each voltage. The measurement time is 13.5 min.

Comment 6: Mention the duration or the applied charge density of the electrolysis experiments performed to obtain the IL-TEM images (Fig. 3a-d). Furthermore, there is no experimental description of the IL-TEM experiments. What kind of cell was used to carry out those experiments?

Response: We are thankful to the reviewer for this valuable comment. Detailed experimental methods for IL-TEM analysis, including duration of electrolysis and reaction system, were added to the corresponding section in the revised manuscript.

Main manuscript Page 17, Line 26: *IL-TEM analysis.* All electrochemical experiments for IL-TEM analysis were performed in a conventional three-electrode system with homemade PEEK cell. Ag/AgCl (3M NaCl) and a graphite rod were used as reference and counter electrodes, respectively. Ag black and Ag-NPs inks were drop-casted on a holey carbon-coated Au grid (Agar scientific, H7 finder grids) and put it on rotating disc electrode (RDE; VSP, Bio-Logic Science Inc.) in a customized holder and the grid was fixed by screwing a PEEK cap to secure electrical contact. The electrochemical measurements were carried out in CO₂-saturated 0.1M KHCO₃, the potential was converted to the RHE scale as following equation:

$$E_{\text{RHE}} = E_{\text{Ag/AgCl}} + 0.209 \text{ V} + 0.059 \text{ V} \times \text{pH}$$

TEM images were collected at the same location of Au grid, and the morphology of the catalysts was compared before and after CO₂RR relying on the applied potentials (-1.0 V_{RHE}) and the reaction time (4 h) while at a rotation speed of 1600 rpm.

Comment 7: The colors of the Metallic-Ag, Ag⁺, and Ag²⁺ mentioned in the caption of Fig. 3g (LFC) do not correspond to the colors observed in the plot.

Response: We are thankful to the reviewer for this valuable comment. Indeed, it was a typo; we corrected caption of Fig. 3g in the revised manuscript.

Fig. 3. Morphology and phase change during the CO₂RR. Identical location transmission electron microscopy (IL-TEM) images of the (a, b) Ag black and (c, d) Ag-NP catalysts (a, c) before and (b, d) after the CO₂RR. (e, f) *In-situ/operando* X-ray adsorption near-edge structure (XANES) spectra at the Ag k-edge for the (e) Ag black and (f) Ag-NP catalysts during the CO₂RR in the MEA-type electrolyzer, and its oxidation state distribution deconvoluted by linear combination fitting (LCF; magenta: metallic Ag, green: Ag⁺, and orange: Ag²⁺).

Comment 8: What was the electrolysis time or applied charge (at the different applied potentials) of the operando synchrotron computed tomographs observed in Fig. 4b?

Response: We are thankful to the reviewer for this valuable comment. The electrolysis time for the *in-situ/operando* CT analysis was 10 min. Related phrase was added to the CT experiment section in the revised manuscript.

Main manuscript Page 19, Line 7: For *in-situ/operando* CT analysis, each potential was applied for 10 min.

Comment 9: On page 12, it is mentioned that “The segmented electrolyte within the electrode surface is shown in Fig. 4d”, I think it is 4b instead of 4d.

Response: We are thankful to the reviewer for this valuable comment. Indeed, it was a typo; we corrected typo in the revised manuscript.

Main manuscript Page 13, Line 12: The segmented electrolyte within the electrode surface is shown in Fig. 4b.

Comment 10: Can the authors comment on the differences between local/not local flooding of the cathode?

Response: We are thankful to the reviewer for this valuable comment. Local flooding refers to a situation where there is a significant decrease in the gas diffusion of reactant gas (CO₂) through electrolyte filling of specific pore or liquid condensation in a localized area or localized liquid film formation. When there are too many local flooding sites, liquid flooding sites merge and the entire large area is flooded, and we expressed this as not local flooding, which is the opposite concept of local flooding.

Comment 11: On page 15, it is mentioned that “Electrochemical CO₂RR single-cell tests. The detailed scheme of an AEM zero-gap-type CO₂ electrolyzer is shown in Supplementary Fig. 3 and Supplementary Fig. 4.” It should be Supplementary Fig. 5 and Supplementary Fig. 6.

Response: We are thankful to the reviewer for this valuable comment. Indeed, it was a typo; we corrected typo in the revised manuscript.

Main manuscript Page 16, Line 10: The detailed scheme of an AEM zero-gap-type CO₂ electrolyzer is shown in Supplementary Fig. 10 and Supplementary Fig. 11.

Comment 12: In the experimental description, it is not clear what “200 cm of humidified CO₂ gas” means. Is it SCCM instead of cm?

Response: We are thankful to the reviewer for this valuable comment. Indeed, it was a typo; we corrected “cm” as “scm” in the revised manuscript.

Main manuscript Page 16, Line 21: To obtain the oxide-derived Ag-based electrodes, a reverse voltage of 3 V was applied to the zero-gap electrolyzer for 2 min. Subsequently, a 0.1 M KHCO₃ solution was supplied to the anode side, while 200 scm of humidified CO₂ gas at 50 °C was inserted into the cathode side.

Comment 13: On page 17, it is mentioned that “the in-situ/operando CT setup is shown in Supplementary Fig. 6 and Supplementary Fig. 7.” It should be Fig. 4a, supplementary figure 14, and supplementary figure 15.

Response: We are thankful to the reviewer for this valuable comment. Indeed, it was a typo; we corrected typo in the revised manuscript.

Main manuscript Page 19, Line 3: Micro-scale *in-situ/operando* CT was performed at the 6C beamline of the PAL; the *in-situ/operando* CT setup is shown in Supplementary Fig. 25 and Supplementary Fig. 26.

Comment 14: The caption of Supplementary Figure 7 is incorrect. (b) CO faradaic efficiency instead of CO partial current density and (c) CO partial current density instead of H₂ partial current density.

Response: We are thankful to the reviewer for this valuable comment. Indeed, it was a typo; we corrected caption of Supplementary Figure 7 in the revised manuscript.

Supplementary Figure 12. CO₂RR performance for Ag-NP catalysts with various TMAH treatment time in 0.1 M KHCO₃ electrolyte. (a) Total current density, (b) CO faradaic efficiency, and (c) CO partial current density.

Reviewer: 2

Recommendation: The present work of Hyung-Suk Oh and co-workers reports the CO₂ electrolysis performance of Ag nanoparticles obtained by a colloidal synthesis followed by partial ligand removal. The importance was given to the effect of the synthesis-derived-ligands on the long term stability for CO₂ in effort to minimize electro-wetting at high current density testing. Most of the figures are clear and there are several operando methods (Synchrotron X-ray CT and XANES) to integrate visual aid for the catalytic activity and stability which are novel to the field of CO₂ electrolysis. Yet, the info provided in the methods and SI are not detailed enough and needs further inputs. There are some logical leaps in the interpretation of the data, especially amongst the samples. At the current state, the conclusions do not seem to present a major advancement in the current state-of-art. Thus, the work is recommended to be go under significant revision after considering the following questions and comments;

Comment 1: The sentence in page 3 claims that

a. “... electrocatalysts with hydrophobic properties have not yet been reported...” .

Several groups have demonstrated the influence of long-chain ligands derived from the colloidal nanoparticle synthesis of Cu, Ag and CuAg such as oleylamine¹ and nitro-containing ligands², displaying a hydrophobic property in CO₂ electrolysis by capacitance and zeta-potential measurements, respectively. Therefore the authors need to rephrase this sentence by assessing the results in the established literature.

b. “... Furthermore, the lipid ligand inhibits Ostwald ripening and sintering of the electrocatalyst during the CO₂RR, thereby facilitating the maintenance of the fine nanoparticle size...”

This phenomenon (catalyst sintering) can be explained by the collapse of the ligand upon its introduction into an aqueous solution. In particular, CO₂ electrolysis is known to be highly alkaline at the local reaction environment, pH > 11 at high current densities > 200 mA/cm².³ In such conditions, the ligands may undergo hydrolysis and the conformal arrangement of Ag-NPs would collapse. This has been demonstrated by ex-situ ligand exchange studies in this article: ACS Catal. 2020, 10, 22, 13468 - 13478

Response: We are thankful to the reviewer for this valuable comment.

a. We modified the phrase based on the references provided by the reviewer.

Main manuscript Page 3, Line 13: However, even with the abovementioned major issues, research in this field has been limited to the microstructural control of electrodes or the introduction of polytetrafluoroethylene (PTFE)^{17,18}; few electrocatalysts with hydrophobic properties have been reported. Several studies introduced the influence of hydrophobic ligand in CO₂RR.^{19,20} However, the real-time observation for the relationship between the behavior of electrolyte and CO₂RR property according to hydrophobicity at three-phase boundary in the device have not yet been reported.

b. OLA is removed by treatment with a high concentration of TMAH solution for several hours. Therefore, it is hardly removed at pH 11~12, which is the local pH formed in the CO₂RR. This can be indirectly confirmed through XRD before and after the reaction. There was no change in particle size in XRD before and after the CO₂RR. Also, there was no change in the crystalline structure. However, in the case of CO₂RR using alkaline electrolyte with high-concentration, OLA may be slowly removed. Related contents added to the revised manuscript.

Supplementary Figure 23. Wide-angle XRD patterns before and after CO₂RR for Ag-NP catalysts.

Main manuscript Page 10, Line 19: As shown in Supplementary Fig. 23, there was no change in crystallinity or particle size for Ag (111) after CO₂RR. In addition, the Ag (110) surface, which is known to exhibit high activity for CO₂RR, also participates in the CO₂RR to increase activity, and there was no degradation after CO₂RR like Ag (111) surface. Although the lipid ligand can be removed by a strong base such as TMAH or a strong reducing agent such as NaBH₄,¹⁹ it does not seem to create a harsh environment, in which the ligand can be degraded, in the zero-gap CO₂ electrolyzer using near the neutral electrolyte.

Comment 2: Colloidal synthesis in OLA-type solutions require hazardous chemicals during the synthesis and the purification steps. On top of that, the yield of the synthesis are usually 30 % and below in terms of grams of metal used. After 50 hours, the selectivity and partial CO current density are not drastically different between the Ag-NPs and commercial Ag-PTFE samples. Could the authors provide further discussion on the novelty in that aspect?

Response: Thank you for bringing this issue to our attention. The purpose of our study was to reveal the change in CO₂RR property according to the increase in the hydrophobicity of the Ag catalyst by observing the behavior of electrolyte at three-phase boundary of the device in real-time. Although there are risks in colloidal synthesis of ligand exchanged metal nanoparticle and disadvantages of low yield. However, since structure and size is very homogeneous, ligand exchange Ag nanoparticles was used as a suitable model catalyst to achieve the purpose of this study.

The degradation rate for Ag-NP and Ag-PTFE catalysts after 50 hours of durability test at high cathodic overpotential was about 26 % and 36 %, respectively. Although the difference in performance reduction rate was not large, the current fluctuation tendency due to the behavior of electrolyte on the surface of catalyst was very large. In device operation, current fluctuation management is one of the important factors to improve stability of zero-gap device (reference: Journal of The Electrochemical Society, 2021, 168, 064507; Electrochemistry Communications, 2007, 9, 497–503). Related expression added in the revised manuscript.

Main manuscript Page 9, Line 3: Although the electrolyte discharged well, the current fluctuation was severe, and there was a slightly greater decrease in performance in relation to that of the Ag-NP catalyst. This is presumably due to the difference in the hydrophobicity homogeneity between the lipid ligand directly attached to Ag and PTFE randomly distributed around Ag. Voltage or current fluctuations in zero-gap electrochemical devices have a detrimental effect on long-term performance

degradation. Therefore, management of current fluctuation is one of important parts of electrochemical device research.

Comment 3: Referring the line in Page 5: “ ... To analyze the hydrophobic properties while maintaining the morphology of the catalyst, by leaving a part of the ligand, physical property analysis was performed using Ag-NP treated with TMAH for 2 h. ...”

a. Figure S2 shows the HR-TEM images of the “OLA-capped Ag-NPs” after 5 h of TMAH treatment to remove OLA, which results with the collapse of the conformal arrangement and the size of Ag-NPs. Hence, an alternative of 2 h TMAH treatment was suggested to preserve the NP stability. However, the evidence for the stability after 2h TMAH treatment is limited – so far, this evidence is limited to three individual Ag particles in Figure S3. Could the authors please provide further evidence with HR-TEM images of the selected process (2h-TMAH), comparable to the images in Figure S2 (or in Fig.1a)? which are the TEM images used to build the histograms of 2h-TMAH sample?

Response: Thank you for your insightful comments. We added TEM images used to measure the particle size distribution of Ag-NP catalyst treated with 2 hours TMAH to the revised manuscript. It can be confirmed from the TEM images that the particle size or shape does not change even after 2 hours of TMAH treatment.

Supplementary Figure 5. HR-TEM images of Ag-NP catalyst with 2 h of TMAH treatment for calculation of particle size distribution and average particle size.

Supplementary Figure 6. HR-TEM image of pristine Ag-NP catalyst

Comment 4: Eventhough Atom Probe Tomography (APT) technique seems to be a strong indicator of the presence of N-containing ligands (such as oleylamine-OLA and tetramethylammonium-TMAH), a conclusive analysis for the presence of the surface groups would be the Fourier Transform Infrared Spectroscopy (FTIR) or the analysis of the colloidal solution by Nuclear Magnetic Resonance (NMR). The N-signal observed by APT could be derived from OLA and/or TMAH, therefore could the authors reach a conclusion at this end and provide stronger evidence about the content of the surface chemistry?

Response: We appreciate the reviewer's helpful comment. We analyzed OLA on the surface of Ag-NPs by NMR analysis. As shown in the Figure below, the singlet signal of 2.5 ppm and the doublet signal of 5.25 ppm are representative peaks of OLA (Reference: Langmuir, 2017, 33, 5456-5463). Through the NMR results and the amine signal of the FT-IR analysis in Figure S4, it can be confirmed that the N signal observed in APT was N of OLA. A description of the NMR analysis method and results was added to the revised manuscript.

Supplementary Figure 7. ¹H NMR spectra of Ag-NP, Oleylamine and TMAH

Main manuscript Page 6, Line 2: Since the N signal might be due to the residue of TMAH, ¹H nuclear magnetic resonance (NMR) analysis was additionally performed. In the spectrum of the Ag-NP were found at near 5.25 ppm and 2.5 ppm that are attributed to the $-\text{CH}=\text{CH}-$ and $-\text{CH}_2\text{NH}_2$ of oleylamine, respectively.²⁶

Main manuscript Page 17, Line 24: The presence of oleylamine was confirmed by ^1H NMR in CDCl_3 using solution state NMR 600 MHz spectrometer (Agilent).

Comment 5: There is a logical leap of using ligands, and the going to partial ligand-exchange/removal and then an oxidation step... What is the motivation for the authors to form an oxide-derived Ag-NPs, instead of the as-synthesized Ag-NPs?

Response: We are thankful to the reviewer for this valuable comment. TMAH treatment was performed to remove an appropriate amount of ligand attached to Ag-NPs and to remove impurity. However, due to concerns that impurity may not be completely removed, additional impurity removal was performed through pre-oxidation treatment with a short time. Related sentence was added to the experimental method section in the revised manuscript.

Main manuscript Page 7, Line 7: Prior to electrochemical measurement, Ag-NP electrocatalysts was oxidized to obtain the oxide-derived electrode for removal of impurities on the surface of Ag-NP.

Main manuscript Page 16, Line 18: The pre-oxidation process was performed to remove impurities remained on the surface of Ag-NP, and there was no change in oleylamine on the Ag-NP surface after pre-oxidation process (Supplementary Fig. 9).

Comment 6: FTIR spectra of the OD-Ag NPs were given Figure S4. The preparation method of this sample for the FTIR analysis is not clear, please specify in the experimental section. The sustanion chemistry may also give signals around $2600 - 3200 \text{ cm}^{-1}$ wavenumbers⁴. After the disassembling the MEA, could the authors be certain that those peaks are only belong to the synthesis ligands of the Ag-NPs? Is there a control sample where commercial Ag-NPs were analyzed by FTIR after an oxidation step in a similar MEA assembly?

Response: Thank you for your insightful comments. We added detailed sample preparation method for FT-IR to the revised manuscript. In the FT-IR experiment, Ag-NP catalyst were sprayed on a double polished silicon substrate and measured after pre-oxidation in a half-cell system. As shown in the Figure below, gas diffusion layer (GDL) did not exhibit a specific signal in the wavenumber range of 2600 to 3200 cm^{-1} . Therefore, the signal for oleylamine can be obtained after MEA test. However, since the transmittance of GDL in that wavenumber range is low compared to double polished silicon substrate, a lot of signal loss will occur using GDL. Therefore, it is easier to observe the change in ligand by TMAH treatment and pre-oxidation adopting half-cell system with double polished silicon substrate.

Supplementary Figure 8. FT-IR spectra ($2600\text{-}3200 \text{ cm}^{-1}$) of the GDL.

Main manuscript Page 7, Line 7: Prior to electrochemical measurement, Ag-NP electrocatalysts was oxidized to obtain the oxide-derived electrode for removal of impurities on the surface of Ag-NP. Fourier transform infrared (FT-IR) spectroscopy was measured after oxidation to observe whether lipid ligand changes due to pre-oxidation process. For measurement under ideal conditions without influence of substrate (Supplementary Fig. 8: FT-IR spectrum of GDL), pre-oxidation process was performed in a half-cell system using a double polished silicon substrate.

Comment 7: Regarding Figure S8,

- a. The Faradaic efficiencies and partial current densities do not match. For ex. at 3.6 V, Ag black shows 70 % FE of H₂ and 500 mA/cm² total cell current, which should be equal to 350 mA/cm² partial current for H₂. Yet, Figure s8c shows only 275 mA/cm². Could you please explain the inconsistency in the efficiency data? Please present a table of results for the tests conducted.
- b. There are not any error bars in any of the data. Is this a single test run? Is there any deviation amongst the GC injections – whether there are multiple injections or not? And are there any duplicate or triplicate tests to minimize the deviation?

Response: We appreciate the reviewer’s helpful comment.

- a. Several figures were miscalculated by confusing selectivity with faradaic efficiency. We added Figures with data errors corrected to the revised manuscript. In addition, a table summarizing the results of each CO₂RR experiment was added to the supplementary information.
- b. We added an error bar to the CO₂RR experiment in the revised manuscript. Error bars were obtained through triplicate tests. The corresponding contents was indicated in the experimental method section. In addition, the several values were corrected in the revised manuscript due to the additional experiments for adding error bars.

Fig. 2. Single-cell performance of the Ag-NP catalyst. (a) Selectivity of CO, and (b) CO partial current density versus applied cell voltage in a zero-gap CO₂ electrolyzer for the Ag black and Ag-NP catalysts. (b, c) Durability test results of the Ag-NP catalyst in the zero-gap CO₂ electrolyzer at (b) 150 mA cm⁻² for 100 h and (c) 3.4 V for 50 h. (d) Durability test result of the Ag black catalyst in the zero-gap CO₂ electrolyzer at 3.4 V for 15 h. Selectivity of CO and H₂ measured during the durability tests. Electrodes of the CO₂ electrolyzer were prepared with 0.3 mg cm⁻² of Ag catalysts on a 10-cm² gas diffusion layer (GDL) on the cathode side. (e) Schematics (low and high magnifications) of the triple phase boundary for the hydrophilic Ag black and hydrophobic Ag-NP catalysts in the CO₂ electrolyzer.

Supplementary Figure 13. CO₂RR performance and HER measured for Ag black and Ag-NP catalysts in 0.1 M KHCO₃ electrolyte. (a) Total current density, (b) H₂ selectivity, (c) H₂ partial current density, and (d) CO faradaic efficiency.

Supplementary Figure 15. CO₂RR performance and HER measured for Ag-PTFE catalysts in 0.1 M KHCO₃ electrolyte. (a) Total current density, (b) CO partial current density, and (c) H₂ partial current density.

Main manuscript Page 2, Line 5: Herein, we found that partially ligand-derived Ag nanoparticles (Ag-NPs) could prevent electrolyte flooding while maintaining catalytic activity for CO₂ electroreduction, exhibiting a high Faradaic efficiency for CO (>90%) and high partial current density (324.6 mA cm⁻²) despite the harsh stability test conditions (3.4 V).

Main manuscript Page 7, Line 33: Furthermore, the maximum partial current density of the Ag-NP catalyst (Ag-NP = 324.6 mA cm⁻²) was almost 25% higher than that of the Ag black catalyst (256.5 mA cm⁻² at 3.4 V).

Main manuscript Page 8, Line 22: A long-term durability test conducted under chronopotentiometric conditions at a constant current of 1.5 A to confirm the feasibility of the Ag-NP catalyst for the sustainable CO₂-to-CO conversion.

Comment 8: In page 7, please clarify or rephrase this statement, as it is inconclusive with the prior sentence. “...Therefore, the partial current density of CO exhibited by the Ag-NP catalyst at low cathodic overpotentials was significantly smaller than the values at high cathodic overpotentials...”

Response: We are thankful to the reviewer for this valuable comment. The statement rephrased in the revised manuscript.

Main manuscript Page 7, Line 32: Therefore, the CO partial current density of Ag-NP and Ag black catalysts was turned around at high cathodic overpotential.

Comment 9: The size of the Ag-black particles need to be clarified, as the authors claim a higher activity for Ag-NPs due to its “moderate size between 5 - 10 nm”. How is this an indication of “...extrinsic effect of the lipid ligand..”, please clarify?

Response: We appreciate your pointing it out. The size effect and the hydrophobicity effect are unrelated, which we described confusingly. Therefore, we modified expression in the revised manuscript. The Ag black catalyst has an average particle size of 20~40 nm. Ag-NP catalyst showed high faradaic efficiency (FE) at high cell voltage, regardless of the average particle size. The difference in maximum partial current density varied depending on the average particle size, and Ag-NP catalyst with an average size of 7.33 nm showed the best performance. The CO₂RR results according to the average particle size of the Ag-NP catalyst were added to the supplementary information of revised manuscript.

Supplementary Figure 14. CO₂RR performance and HER measured for Ag-NP catalysts with various particle size in 0.1 M KHCO₃ electrolyte. (a) Total current density, (b) CO faradaic efficiency, and (c) CO partial current density.

Main manuscript Page 7, Line 33: Furthermore, the maximum partial current density of the Ag-NP catalyst (Ag-NP = 324.6 mA cm⁻²) was almost 25% higher than that of the Ag black catalyst (256.5 mA cm⁻² at 3.4 V). These results indicate an extrinsic effect of the lipid ligand on the FE_{CO} maintenance of the Ag-NP catalyst. In addition, the CO₂RR performance of Ag-NP catalysts with

various particle size was confirmed. As shown in Supplementary Fig. 14, Ag-NP catalyst with average particle size of 7.33 nm exhibited the highest current density. This is consistent with the results of a previous study that Ag-NPs with a moderate size, between 5 and 10 nm, exhibit the highest CO₂RR activity³⁰. According to the result for CO₂RR performance, Ag-NP catalyst with an average particle size of 7.33 nm was used for various analyzes to be discussed below.

Comment 10: In Figure S9,

a. The partial current densities do not add up to the total value. For ex. at 3.4 V, the total j is around 300 mA/cm², while the j_{CO} is between 220 - 250 mA/cm² and j_{H₂} is almost zero, please clarify the breakdown of the calculation, preferably with table of results?

Response: We appreciate reviewer's valuable comment. In several Figures, CO selectivity on the y-axis was incorrectly described as Faradaic efficiency. We added Figures with corrected y-axis captions to the revised manuscript. We also added a table of CO₂RR results for all catalysts in the revised manuscript.

Supplementary Table 1. Summary of experimental result for AEM zero-gap-type CO₂ electrolyzer.

Ag catalyst	Cell potential (V)	Total current density (mA cm ⁻²)	CO F.E. (%)	Partial Current density (mA cm ⁻²)	CO selectivity (%)
Ag black	2.6	92.1	91.8	84.5	98.9
	2.8	177.0	89.2	157.8	98.7
	3.0	282.2	82.0	231.3	97.9
	3.2	390.9	65.6	256.5	80.3
	3.4	506.7	27.3	138.5	33.1
	3.6	32.9	84.7	27.9	97.0
Ag-NP	2.6	71.0	94.2	66.9	98.1
	2.8	135.2	95.9	129.7	97.9
	3.0	228.6	97.0	221.8	97.0
	3.2	354.6	91.5	324.6	95.1
	3.4	455.9	62.3	284.1	72.5
	3.6	27.0	92.2	24.9	96.3
Ag-PTFE (10 wt%)	2.6	67.5	97.2	65.6	97.5
	2.8	141.8	96.7	137.1	97.8
	3.0	242.8	90.5	219.6	97.9
	3.2	328.2	83.0	272.5	97.3

	3.4	395.1	43.4	171.3	53.2
	3.6	27.3	95.2	26.0	97.3
	2.6	64.5	98.0	63.2	98.5
	2.8	133.7	96.9	129.6	98.5
Ag-PTFE (20 wt%)	3.0	219.1	90.9	199.2	98.3
	3.2	284.7	88.4	251.7	98.1
	3.4	330.0	64.5	212.8	83.1
	3.6	330.0	64.5	212.8	83.1

Main manuscript Page 17, Line 6: The experimental results of AEM zero-gap-type CO₂ electrolyzer were summarized in Supplementary Table 1.

Comment 11: Regarding line “ .. This is due to a decrease in the active surface area at the expense of suppressing the flooding phenomenon at a high cell voltage...”

- Are there any ECSA measurement to support this claim?
- For Ag-PTFE, the total cell current is less compared to Ag-NPs, which points to a higher resistance under fixed voltage testing. Hence, the authors need to consider the electrical resistance contribution of the PTFE additive instead.

Response: We appreciate your pointing it out.

- We measured CV for Ag catalysts in the non-Faradaic potential range. As shown in the Figure below, Ag black catalyst showed the largest active surface area compared to other catalysts, which was consistent with the total current density result of the CO₂RR experiment.

Supplementary Figure 16. CV measured for (a) Ag black, (b) Ag-NP, (c) Ag-PTFE 10%, and (d) Ag-PTFE 20% catalysts in non-Faradaic potential range at various scan rate. (e) Linear fitting of the capacitive currents versus CV scan rates.

Main manuscript Page 8, Line 16: This is due to a decrease in the active surface area at the expense of suppressing the flooding phenomenon at a high cell voltage (Supplementary Fig. 16: CV in non-Faradaic potential range for ECSA measurement).

- b. We added impedance result for Ag-NP and Ag-PTFE catalysts to the supplementary information of revised manuscript. The ohmic resistance of the Ag-NP and Ag-PTFE 10% catalysts did not differ significantly. Therefore, the effect of the ohmic resistance in CO₂RR current density between Ag-NP and Ag-PTFE catalysts was negligible. Related phrase added to the revised manuscript.

Supplementary Figure 17. Electrochemical impedance spectra for Ag black, Ag-NP, Ag-PTFE 10%, and Ag-PTFE 20% catalysts at a cell voltage of -3 V.

Main manuscript Page 8, Line 18: In addition, there was an effect of increasing ohmic resistance due to PTFE (Supplementary Fig. 17: Impedance spectra at cell voltage of -3 V). As the amount of PTFE increase, the ohmic resistance increases and affects the total current density.

Comment 12: In Figure 2,

- a. In Fig 2b, the cell voltage shows a steady increase 80th hour from 3.1 V to ca. 3.5 V, and then experiences a sudden drop back to 3 V. Could the authors please explain why such a change has occurred?
- b. In Figure 2c-d & Fig S10, did the authors facilitated water removal externally by flushing with a liquid or inert gas media in any of the tests, at any time point? If yes, please specify the details in the experimental section.
- c. In comparison of the j_{CO} performance of Ag-PTFE vs. OD-Ag-NPs treated with TMAH, the starting and ending partial CO current densities are identical (350 and 225 mA/cm²) along the 50 hours stability test. Yet, the water management property of the Ag-PTFE vs. OD-Ag-NPs treated with TMAH are different regarding the observed current spikes (150 vs 200 mA/cm² min. point). The authors need to acknowledge the similarity in the performance (y-axis is over-stretched in the Fig S10 in comparison to Fig. 2c!) but they need to provide further evidence on, what sort of mechanism derives the difference in the local water management? (the last sentence of the paragraph in page 8 needs further evidence / discussion)

Response: We are thankful to the reviewer for this valuable comment.

- a. The reason of the potential recovery is not clear. However, it is possible to predict that the potential recovery during the durability test is derived by the following reasons; 1) changes in cation concentration due to electrolyte replacement, 2) increase in HER, and 3) water discharge from the electrode to cathode outlet. In the case of our experiment, it is assumed that the potential recovery as the electrode solution gradually accumulated on the electrode was discharged through the cathode outlet trap.
- b. We did not remove the electrolyte through a liquid or inert gas. Since the purpose of our study was to determine how much the flooding phenomenon affecting durability was improved via the hydrophobicity of the cathode, a mobile phase, such as Ar or N₂ gas, was not used to remove electrolyte during the reaction.
- c. We corrected the sentence referring to the difference in durability between Ag-NP and Ag-PTFE catalysts. The degradation rate of current density for Ag-NP and Ag-PTFE catalysts after 50 hours of durability test at high cathodic overpotential was about 26 % and 36 %, respectively. As we speculate, the Ag-PTFE electrode was a physically mixture of Ag and PTFE, so it is not completely evenly mixed in particle units. Therefore, it is assumed that there was a hydrophobic area due to PTFE and a relatively hydrophilic area with only Ag particles, so current fluctuation occurred more severely than Ag-NP catalyst. We added EDS mapping images for the Ag-NP and Ag-PTFE catalysts to the revised manuscript.

Supplementary Figure 19. EDS mapping images of (a) Ag-NP and (b) Ag-PTFE electrodes.

Main manuscript Page 9, Line 5: This is presumably due to the difference in the hydrophobicity homogeneity between the lipid ligand directly attached to Ag and PTFE randomly distributed around Ag (Supplementary Fig. 19: EDS mapping for Ag-NP and Ag-PTFE electrodes for confirmation of distribution of lipid ligand and PTFE, respectively).

Comment 13: About the IL-TEM,

- The details of the sample preparation for the IL-TEM is not mentioned at any part of the manuscript or SI. Please clarify how are those steps following the disassembly of the MEA and preparing a TEM specimen for the post-mortem analysis?
- Please provide information about the cell configuration and electrodes used for the ex-situ analysis here.

Response: Thank you for bringing this issue to our attention. We added experimental method for IL-TEM, including grid preparation, to the revised manuscript. In addition, cell configuration and electrode preparation methods for ex-situ analyzes were added to the “Methods” section of the revised manuscript.

Main manuscript Page 17, Line 26: *IL-TEM analysis.* All electrochemical experiments for IL-TEM

analysis were performed in a conventional three-electrode system with homemade PEEK cell. Ag/AgCl (3M NaCl) and a graphite rod were used as reference and counter electrodes, respectively. Ag black and Ag-NPs inks were drop-casted on a holey carbon-coated Au grid (Agar scientific, H7 finder grids) and put it on rotating disc electrode (RDE; VSP, Bio-Logic Science Inc.) in a customized holder and the grid was fixed by screwing a PEEK cap to secure electrical contact. The electrochemical measurements were carried out in CO₂-saturated 0.1M KHCO₃, the potential was converted to the RHE scale as following equation:

$$E_{\text{RHE}} = E_{\text{Ag/AgCl}} + 0.209 \text{ V} + 0.059 \text{ V} \times \text{pH}$$

TEM images were collected at the same location of Au grid, and the morphology of the catalysts was compared before and after CO₂RR relying on the applied potentials (-1.0 V_{RHE}) and the reaction time (4 h) while at a rotation speed of 1600 rpm.

Main manuscript Page 17, Line 11: Wide-angle XRD (Bruker Bruker D8 Advance instrument, Cu K α radiation) was employed to determine the crystal structure of the Ag-NP catalyst. After the AEM zero-gap-type CO₂ electrolyzer test, GDL was measured to observe the change in crystallinity after CO₂RR. A drop-shape analyzer (Kruss DSA 100) was used to measure the contact angle of deionized water at each CO₂RR potential. WCA was measured after AEM zero-gap-type CO₂ electrolyzer test for each cell voltage. Cathode electrode for wide-angle XRD and WCA analysis was fabricated the same procedure as single-cell test.

Main manuscript Page 17, Line 20: Pre-oxidation and CO₂RR for *ex-situ* FT-IR measurement were performed using VSP potentiostat (Bio-Logic) in a conventional three-electrode system with homemade polyether ether ketone (PEEK) cell equipped a Ag/AgCl (3M NaCl) and a graphite rod as the reference and counter electrodes, respectively.

Comment 14: In page 10, regarding line “... They expose the high-crystallinity (111) plane, inducing excellent catalytic activity toward the CO₂RR under cation-assisted conditions^{32,33}, enhancing the CO₂RR activity without cathodic corrosion...” This postulation needs further evidence such as; the authors need to justify this postulation with a before and after sample analysis with XRD pattern. The Ag (110) facet was shown by both experimental and theoretical work to be more active than the very stable (111) facet⁵.

Response: We appreciate reviewer’s valuable comment. Face-centered cubic crystals have the highest density of facets in the {111} direction. Therefore, the presence of CO₂RR activity on the (111), which is the most preferred plane, was expressed in the discussion. In the XRD spectra before and after the CO₂RR, it is possible to confirm that the (111) plane was the preferred plane compared to other planes. Furthermore, there was no change in particle size and crystallinity before and after the CO₂RR. However, the (110) plane also existed, and it can also participate in the CO₂RR based on high activity. We added related contents to the revised manuscript based on the references recommended by reviewer.

Supplementary Figure 23. Wide-angle XRD patterns before and after CO₂RR for Ag-NP catalysts.

Main manuscript Page 10, Line 19: As shown in Supplementary Fig. 23, there was no change in crystallinity or particle size for Ag (111) after CO₂RR. In addition, the Ag (110) surface, which is known to exhibit high activity for CO₂RR, also participates in the CO₂RR to increase activity, and there was no degradation after CO₂RR like Ag (111) surface.

Comment 15: The part about the XANES spectra in page 10, 2nd paragraph needs further discussion. It is advised to provide a brief information about the Operando X-ray absorption spectroscopy for the broader audience, as in Ref 5. The parameters such as the sample tilt needs to be addressed as it effects the penetration depth of the X-rays from 2 nm to 12 μm at 0 to 45 angles.

a. The analysis of the results are not easy to follow such as the meaning of LCF is not explained. For ex. the quantification methods and the data analysis details needs to be reported in the SI and/ or the fitting results of the reference metal, metal-oxides and samples of Fig 3e-f need to be reported in a table.

(see example <https://www.rsc.org/suppdata/c8/ta/c8ta10412c/c8ta10412c1.pdf>)

b. Could the authors elaborate on whether the beam alone, at the intensity used & timescale of the reaction, was capable of reducing the silver or not?

Response: We are thankful to the reviewer for this valuable comment.

a. We added detailed information about the hard XAS analysis system and quantification method to the revised manuscript. In addition, XANES spectra of metallic Ag, Ag⁺ and Ag²⁺ required for LCF analysis were added to the supplementary information, and the results for each fitting were also tabulated.

Supplementary Figure 24. XANES spectra at the Ag k-edge for the Ag foil, Ag(I) oxide, and Ag(II) oxide references.

Supplementary Table 1. XANES fitting results of Ag black and Ag-NP catalysts

	Ag black			Ag-NP		
	Ag ⁰	Ag ¹⁺	Ag ²⁺	Ag ⁰	Ag ¹⁺	Ag ²⁺
ex-situ	0.886	0.112	0.020	0.823	0.159	0.018
OD	0.383	0.488	0.129	0.513	0.392	0.950
2.6 V	0.672	0.265	0.063	0.655	0.304	0.041
3.4 V	0.772	0.176	0.052	0.700	0.262	0.038

Main manuscript Page 10, Line 29: The penetration depth of XANES analysis is about 3 μm for Ag.⁴⁰ Therefore, bulk signals are detected together with information of surface. However, nanoparticles can be used for surface analysis because of the low interference of bulk signal.

Main manuscript Page 18, Line 26: The photon flux of incoming hard X-ray was about 1×10^{11} photons per second and the nominal beam size was 1 mm x 1 mm. The reference spectra of the Ag foil, Ag(I) oxide and Ag(II) oxide were obtained by transmission mode in Ar filled chamber under ambient pressure and room temperature (Supplementary Fig. 24: XANES spectra of Ag foil, Ag(I) oxide and Ag(II) oxide).

Main manuscript Page 19, Line 1: A description of the LCF method was described in Supplementary Statement 1 and the fitting results were tabulated in Supplementary Table 1.

Supplementary Statement 1. XANES analysis: Linear combination fitting (LCF)

The general approach to XANES analysis is to treat measured XANES data as a linear mixture of the XANES spectra of reference components, such as metal foil and bulk oxide. This fitting method on the

assumption that the XANES signal of atoms collection is the linear sum of the XANES from individual components, which is valid under all conditions except the harsh conditions. In this sense, LCF is a useful approach to XANES analysis, and is generally very easy to perform. Sensitivity can be somewhat limited if done carefully, but can also be quite robust fitting method.

- b. Metal reduction may occur during hard XAS measurement if the beam flux is large (reference: The Canadian Journal of Chemical Engineering, 2022, 100, 3–22; The Journal of Physical Chemistry C, 2021, 125, 11048–11057). Reduction can be conspicuous when exposed over tens of minutes. However, since XANES analysis was performed for a short time (~ 3 min), the effect of artifact due to beam damage was considered to be extremely small.

Comment 16: WCA of 3.4 V is shown in Fig 4c but its value is missing in the chart of Fig 4d, instead 3.8 V was shown. Is this intentional or typo error? What is the contact angle of the electrode after 3.4 V?

Response: We are thankful to the reviewer for this valuable comment. Indeed, it was a typo; we corrected typo of Figure 4d in the revised manuscript. WCA after CO₂RR at 3.4V of Ag black and Ag-NP catalysts was measured as 80.53° and 97.41°, respectively.

Fig. 4. Visual analyses of the influence of electrode hydrophobicity on the CO₂RR. (a) A schematic representation of the zero-gap CO₂ electrolyzer used for the *in-situ/operando* synchrotron computed tomography (CT) and cross-section tomography of the Ag-NP cathode. (b) High-magnification synchrotron tomographs of the 3D structure of the Ag black and Ag-NP cathodes at various applied cell voltages (blue: electrolyte). (c) Water contact angle (WCA) images for the Ag black and Ag-NP cathodes at the initial state, 2.8 V, and 3.4 V. (d) WCA and water volume fraction versus the applied cell voltage in the zero-gap CO₂ electrolyzer for the Ag black and Ag-NP catalysts. The water volume fraction in the electrodes at each applied cell voltage was estimated from the tomographs.

REVIEWER COMMENTS

Reviewer #1 (Remarks to the Author):

The authors have thoroughly replied to the reviewer's questions, improving the manuscript. However, it needs to be clearer how the TEM analysis was done. First, the performance of the Ag-black and Ag-NPs was tested in a zero-gap CO₂ electrolyzer system, and the IL-TEM experiments were done using an aqueous electrolyte (CO₂-saturated 0.1 M KHCO₃). Even though the authors mentioned that the applied potential for those experiments was -1 V vs RHE, no information about the reached current density at that potential was included. Nevertheless, it is well-known that the current densities in aqueous electrolytes are limited due to the low solubility of CO₂ in those electrolytes. The reaction environment is not the same in the zero-gap CO₂ electrolyzer as when an aqueous electrolyte is used. Therefore, the observed changes in the Ag-black and Ag-NPs presented in Fig. 3a-d might not represent what happens in a zero-gap electrolyzer, especially when high current densities are reached (more than 300 mA/cm²). It will be helpful that the authors clearly mention why they analyze the catalyst changes in a system that does not represent entirely the same conditions as the zero-gap electrolyzer.

On the other hand, nothing is mentioned about the experimental part of how the Supplementary Figure 20-22 images were obtained. However, if those images were acquired after 50 h in the zero gap at 3.4 V, it would be better to include them in the main manuscript because they are more representative of what is happening to the catalyst morphology when they are tested in zero-gap electrolyzer. For example, suppose the images of the Figure 3d and Supplementary Figure 21 are compared. In that case, it is noticeable that the effects of cathodic corrosion are stronger (larger aggregates and more splitting of the Ag nanoparticles are observed) when the experiments are performed using a zero-gap electrolyzer (Supplementary Figure 21), where higher current densities are attained. Conversely, the changes after the CO₂ electroreduction reaction in the aqueous electrolyte (Fig. 3c-d) are lesser, and this is most likely because of the low current density reached at -1 V vs RHE in the aqueous 0.1 M KHCO₃.

The values presented in the Supplementary Table 1 do not match at all with the data presented in the Figure 2a, Supplementary Figure 13, and Supplementary Figure 15. The authors should check those values and mention to which Figures they are related. For example, with the first three catalysts (Ag black, Ag-NP, and Ag-PTFE-10wt%), the total current density values shown in the mentioned table are increasing, but in the last applied potential, there is a sudden decrease to ~30 mA cm⁻², however, that is not observed in the Supplementary Figure 13a nor in the Supplementary Figure 15a, where the total current density is continuously increasing with an increasing cell potential. Moreover, the authors present in the Supplementary Table 1 a column with the CO F.E. (%) and CO selectivity (%) values. Can the authors explain the difference between those two values? According to my understanding, those values represent the same. Besides, the CO F.E. values in the

Supplementary Table 1 of the Ag black, Ag-NP, and Ag-PTFE-10wt% always decrease at 3.4 V, but they increase at 3.6 V, which does not represent what is shown in the Figure 2a and the Supplementary Figure 13d. Furthermore, it needs to be mentioned whether the column of partial current density values of the Supplementary Table 1 corresponds to CO or H₂.

The Figure 2e represents a scheme of what is happening on the surface of the electrode during the CO₂ electroreduction when a Ag-black and a Ag-NP catalysts are used. Still, no comments are mentioned about this figure in the main manuscript.

There is no caption for the Fig. 3g.

Include in the main text of the manuscript what the Supplementary Figure 7 represents.

Which experimental conditions were applied for the XRD characterization of the Ag-NP after the CO₂RR in the Supplementary Figure 23?

I recommend publishing the manuscript because it is of interest to the readership of Nature Communications only if the previous remarks are addressed in the main manuscript.

Reviewer #2 (Remarks to the Author):

In overall, the article has improved after the 1st review, yet there are details in the data that needs to be revised and remarks to be addressed. Besides, the grammar needs to be improved to match the standards of the publisher. The authors are kindly requested to give answer to a few questions below:

1. What are the conditions of the “heat treatment”, temperature, gas type, and heat/cool-ramp-dwell times employed in order to decompose the impurities remained after the ligand-exchange step? (following Comment 5 response)
2. What is the protocol for the “pre-oxidation step”, could you please explain in detail? (following the Comment 6 response)

3. The authors need to be careful when they talk about the tests results of the “Ag-NPs before and after CO₂RR” as the audience may think that these are the “Ag-GDE samples taken from the MEA after the CO₂RR” – which is not the case... Proxy experiments are useful, and required for further analysis but still; this needs to be expressed clearly to the reader that they are not the exact samples.

a. The sample results for XRD, IL-TEM, EDS-Mapping, XANES, IR-Spectra, WCA are obtained on a different substrate in a different cell geometry – i.e. conventional batch-cells. Even though the applied potential may be the same (-1V_{rhe}), the current density hence the local-environment is totally different ($j_{total} < 30 \text{ mA/cm}^2$ vs. $>300 \text{ mA/cm}^2$ for ex.).

4. Finally, the main message of the article needs to be clear in the abstract and title. If the flooding is investigated using in-situ techniques, then use of Ag particles with and without PTFE binder would suffice given the drastic change between the pristine vs. PTFE mixed catalyst layers.

Yet, the use of Ag-NPs (assisted with a ligand-exchange) is compared with Ag-PTFE mixture GDE. I still believe that, in this article, the novelty needs to be pronounced more effectively by postulating the phenomenon, which (to my understanding so far) is:

- The distribution of PTFE around commercial Ag particles vs. local-coordination of OLA/TMAH ligands showed different hydrophobic behavior – locally.
- It is known that the local-reaction environment at higher current densities ($> 200 \text{ mA/cm}^2$) could expedite the diffusion-limited region to several hundreds of nanometers, demanding a greater mass transfer rate of CO₂. Hence, we need to modify the surface within a few-hundred nanometer range to push the activity and preserve the stability.
- The three-phase-interface is preserved and pronounced better in the ligand-driven system. The interconnected ligand and nanoparticles can serve as the tunnels for the transfer of reactants and product, i.e. CO₂ and CO, respectively.
- In comparison to the randomly distributed PTFE commonly used in literature, the results here are quite sufficient to elaborate the benefit of an advanced interface engineering to push the activity and preserve the stability.
- (Your “ response c” for the Comment 12 is a good postulation, which could be inserted where you lay-out your discussion in the main article)

Reviewer #3 (Remarks to the Author):

Here, Ko et al. presented their work regarding improving the stability of Ag nanoparticles towards selective CO₂RR with hydrophobic ligands. In general, I find the results from the author's experiments to be convincing of the mechanisms they had proposed. However, there are also several

issues that I believe the authors need to address before the paper can be considered for publication, as I describe below.

Figure 1

The authors should provide low-magnification images of the nanoparticles after two hours of TMAH treatment in the supplementary information to support their claim of particle uniformity.

What is the difference between the two slices shown in Figure 1d? Are they from different locations in the tip? It is not clear to me what I am supposed to infer from the two slices. Why do the authors extract the profiles in 1(e) from an area which does not obviously show a silver nanoparticle?

The authors should provide scale bars from the slices.

Also related to the Figure 1(d). In line 134-135, the authors state that "Fig. 1c–d show a needle-shaped tip and its reconstructed 3D atom map containing several Ag-NPs and an iso-concentration surface of carbon."

What does iso-concentration surface of carbon mean?

In several parts of the manuscript, the authors state that the ligands are attached to the edge of nanocrystals.

Line 128-129 Therefore, the CO₂RR selectivity is expected to improve upon attachment of the ligand to the edge of the nanocrystal.

Line 174-175 The Ag-NP catalyst subjected to 2 h of TMAH treatment showed the highest performance because an appropriate amount of ligand attached on the surface, and there was no change in the icosahedron morphology while maintaining the active site of Ag.

Line 351-354 "Owing to the influence of the lipid ligands occupying the edge sites of the Ag NPs, cathodic corrosion and subsequent carbonate-ion adsorption at high cathodic overpotentials occurred to a comparatively lesser extent in the Ag-NP cathode than in the Ag black cathode."

Here, I don't see any experimental evidence of edge attachment or references to previous work that show edge attachment. The authors should clarify.

Line 165-166 "Although the electrode is oxide-derived, the lipid ligand is well maintained."

It is not clear to me what is expected or unexpected from the oxide-derived catalysts. Do the authors expect them to restructure? The authors should clarify.

Line 173-175 "The Ag-NP catalyst subjected to 2 h of TMAH treatment showed the highest performance because an appropriate amount of ligand attached on the surface, and there was no change in the icosahedron morphology while maintaining the active site of Ag."

How do the authors know there is no change in the catalysis morphology during electrolysis at this point of the paper? Also, how do the authors know where is the active site of Ag?

Line 180 What is "Ag black"? I find no description of how this "Ag black" is prepared and its associated characterization.

Line 243-244 "The reaction was conducted at -1.0 V vs. RHE, near the highest partial current density."

How did the authors determine that -1.0 V vs. RHE in the IL TEM cell replicates the behavior found in their electrolyzer? Here, the reaction environment and current densities are likely to be quite different between the two setups and may alter the morphological changes. The authors need to show that they can establish equivalence. Otherwise, they need to preface their discussion with a disclaimer that the two systems are not exactly equivalent.

Other issues that they need to clarify regarding the IL-TEM include whether they see bubble formation on the working electrode of the IL-TEM cell and if there is any impact of the TEM grid material. Au is also known to be selective towards CO.

Line 254-255 "This difference could be attributed to the rarely degradable lipid ligands that are attached to the edge of the nanoparticles."

What does "rarely degradable" mean? Also, see above comment regarding edge attachment.

Minor Comments"

Supplementary Figure 5. The authors state in the caption of Figure 5 that Supplementary Figure Image 5-5 is used in Figure 1(a).

Reviewer #3 (Remarks to the Author):

Here, Ko et al. presented their work regarding improving the stability of Ag nanoparticles towards selective CO₂RR with hydrophobic ligands. In general, I find the results from the author's experiments to be convincing of the mechanisms they had proposed. However, there are also several issues that I believe the authors need to address before the paper can be considered for publication, as I describe below.

Comment 1: The authors should provide low-magnification images of the nanoparticles after two hours of TMAH treatment in the supplementary information to support their claim of particle uniformity.

What is the difference between the two slices shown in Figure 1d? Are they from different locations in the tip? It is not clear to me what I am supposed to infer from the two slices. Why do the authors extract the profiles in 1(e) from an area which does not obviously show a silver nanoparticle?

Response: We appreciate the reviewer's helpful comment. We re-obtained TEM images of Ag-NP with TMAH treatment of 2h and 5h. For the existing TEM images, TEM analysis was performed after TMAH treatment of pristine Ag-NP coated TEM grid, but the newly measured TEM images were obtained by applying Ag-NP samples sonicated in a TMAH solution to TEM grids. Additionally, the particle size distribution for obtained TEM images for Ag-NP with 2h of TMAH treatment was recalculated. We added low-magnification TEM images of TMAH-treated Ag-NP sample.

The two slices shown in Figure 1d are the same region of the APT tip. The left side is a slice marked with high contrast of Ag to show the location of Ag particles, and the right side is a slice marked with the same level of darkness throughout. Therefore, the profile in Figure 1e is the data obtained from the line profile in the area with the isolated Ag nanoparticle. Such information was added to the caption of Figure 1 in the revised manuscript. Please check the new Figure 1 and Supplementary Figure 1 on Page R3 and R8, respectively.

Comment 2: The authors should provide scale bars from the slices.

Also related to the Figure 1(d). **In line 134-135**, the authors state that "Fig. 1c–d show a needle-shaped tip and its reconstructed 3D atom map containing several Ag-NPs and an iso-concentration surface of carbon."

What does iso-concentration surface of carbon mean?

Response: Thank you for this helpful comment. We added scale bars to the slices in Figure 1c–d in the revised manuscript. The iso-concentration surfaces are commonly used to delineate phases in atomic probe datasets. These surfaces then provide spatial and compositional reference for proximity histograms, density and volume of atoms within a multiphase system. Please check the new Figure 1 on Page R3.

Comment 3: In several parts of the manuscript, the authors state that the ligands are attached to the edge of nanocrystals.

Line 128-129 Therefore, the CO₂RR selectivity is expected to improve upon attachment of the ligand to the edge of the nanocrystal.

Line 174-175 The Ag-NP catalyst subjected to 2 h of TMAH treatment showed the highest performance

because an appropriate amount of ligand attached on the surface, and there was no change in the icosahedron morphology while maintaining the active site of Ag.

Line 351-354 "Owing to the influence of the lipid ligands occupying the edge sites of the Ag NPs, cathodic corrosion and subsequent carbonate-ion adsorption at high cathodic overpotentials occurred to a comparatively lesser extent in the Ag-NP cathode than in the Ag black cathode."

Here, I don't see any experimental evidence of edge attachment or references to previous work that show edge attachment. The authors should clarify.

Response: We appreciate the reviewer's helpful comment. We used the corner site and edge site confusedly. The amidogen groups of oleylamine can ligand to the low-coordinated sites, especially corner sites, to minimize the surface energy. Thus, when the oleylamine ligands coordinate on the surface of the Ag-NPs, they are more likely to occupy the corner sites. We corrected expressions and added a reference to the binding of the ligand to the corner site of metal particles to the revised manuscript.

Reference: Chem. Commun., 2020, 56, 7021

Main manuscript Page 5, Line 24: To minimize the surface energy, the amidogen groups of the oleylamine ligands were attached to low-coordinated sites, especially the corner sites.²⁸ When the ligands coordinated on the Ag-NP surface, they were more likely to occupy the corner sites. Therefore, the CO₂RR selectivity was expected to improve upon attachment of the ligands to the corner sites of the nanocrystals.

Main manuscript Page 12, Line 13: This difference could be attributed to the remaining lipid ligands that were attached to the corner site of the nanoparticles.

Comment 4: Line 165-166 "Although the electrode is oxide-derived, the lipid ligand is well maintained."

It is not clear to me what is expected or unexpected from the oxide-derived catalysts. Do the authors expect them to restructure? The authors should clarify.

Response: We are thankful to the reviewer for this valuable comment. The expression in the Line 165-166 means that there was no structural degradation of the catalyst even though pre-oxidation treatment had been performed. We modified this expression more clearly in the revised manuscript.

Main manuscript Page 8, Line 13: Although the electrode was oxide-derived, the lipid ligand was not removed from the electrode surface. This indirectly suggests that there was little structural change in Ag-NPs.

Comment 5: Line 173-175 "The Ag-NP catalyst subjected to 2 h of TMAH treatment showed the highest performance because an appropriate amount of ligand attached on the surface, and there was no change in the icosahedron morphology while maintaining the active site of Ag."

How do the authors know there is no change in the catalysis morphology during electrolysis at this point of the paper? Also, how do the authors know where is the active site of Ag?

Response: Thank you for your insightful comments. Through the IL-TEM experiment in Figure 3, it was observed that the morphology change of particles was very small even when potential was applied. In addition, relatively little particle agglomeration was observed even after durability experiments, at a high cell potential of 3.4 V. However, since the change in morphology had not been verified at the point of Figure 2, the expression seems to be erroneous. Therefore, we rephrased the sentences in the revised manuscript.

Main manuscript Page 8, Line 23: TMAH treatment for 2 h was predicted to be the optimal condition to maintain the icosahedral shape (Fig. 1d and Supplementary Fig. 5) while increasing the number of active sites of Ag through the appropriate amount of ligand removal.

Comment 6: Line 180 What is “Ag black”? I find no description of how this "Ag black" is prepared and its associated characterization.

Response: Thank you for bringing this issue to our attention. The black catalyst is a fine powder of metal. The name of black is due to its black color.

Comment 7: Line 243-244 "The reaction was conducted at -1.0 V vs. RHE, near the highest partial current density."

How did the authors determine that -1.0 V vs. RHE in the IL TEM cell replicates the behavior found in their electrolyzer? Here, the reaction environment and current densities are likely to be quite different between the two setups and may alter the morphological changes. The authors need to show that they can establish equivalence. Otherwise, they need to preface their discussion with a disclaimer that the two systems are not exactly equivalent.

Other issues that they need to clarify regarding the IL-TEM include whether they see bubble formation on the working electrode of the IL-TEM cell and if there is any impact of the TEM grid material. Au is also known to be selective towards CO.

Response: We appreciate the reviewer’s helpful comment. When the potential applied to the cathode was measured in the zero-gap electrolyzer for the Ag black catalyst, about -1 V vs RHE (-1.610 V vs Ag/AgCl) was applied at about 100 mA cm⁻², which was almost similar to the applied potential in the IL-TEM experiment. Therefore, the effect of the under potential can be observed to some extent through the IL-TEM experiment, despite the slightly difference reaction environments between the zero-gap electrolyzer and aqueous electrolyte. IL-TEM analysis is not affected by gas bubbles because it analyzes morphology changes before and after Rotation Disk electrode (RDE) experiments. And since the surface area of Au grid is very small compared to Ag black and Ag-NP catalysts, the amount of reaction is negligible at -1.0 V vs RHE. Such discussion and relevant experiment results were added to the revised manuscript.

Reference: 1. Chem. Commun., 2014, 50, 11143-11146, 2. Nano Res. 2019, 12, 2330-2334, 3.

Supplementary Table 1. Cathode potential in a zero-gap electrolyzer experiment for Ag black catalyst.

Total Cell voltage	Current density	Anode Voltage	Cathode Voltage
(V)	(mA cm⁻²)	(V)	(V)
2.6	51.1277	1.07128	1.52872
2.8	97.2284	1.1166	1.6834
3.0	174.987	1.16315	1.83685
3.2	279.084	1.20479	1.99521
3.4	355.273	1.25557	2.14443
3.6	417.298	1.30137	2.29863

Main manuscript Page 11, Line 12: The reaction was conducted at -1.0 V vs. RHE (-1.610 V vs. Ag/AgCl), near the partial current density of 100 mA cm⁻² in a zero-gap electrolyzer experiment (Supplementary Table 1: the cathode potential in a zero-gap electrolyzer experiment with a Ag black catalyst obtained by adopting a reference electrode). Since many bubbles were generated at a high current density, morphology changes were observed after a long time (4 h) at a relatively low current density. Therefore, despite the slightly different reaction environments between the zero-gap electrolyzer and the aqueous electrolyte, the effect of the CO₂RR potential on the morphology of the catalysts was observed to some extent through the IL-TEM experiment.

Supplementary Figure 29. Chronoamperometric response recorded in 0.1 M KHCO₃ solution at 1.0 V vs RHE for bare Au grid.

Main manuscript Page 20, Line 11: Supplementary Fig. 29 shows the CO₂RR performance of the bare Au grid, which was negligible compared to the CO₂RR performance of the Ag-NP and Ag black-coated Au grids.

Comment 8: Line 254-255 "This difference could be attributed to the rarely degradable lipid ligands that are attached to the edge of the nanoparticles."

What does "rarely degradable" mean? Also, see above comment regarding edge attachment.

Response: We are thankful to the reviewer for this valuable comment. The "rarely degradable" expressed a phenomenon in which the ligands are desorbed on the Ag-NPs or decomposed. In other words, it expressed all the phenomena in which ligands disappear from Ag-NPs. We modified such sentences to be more intuitive in the revised manuscript. Also, the response to 'edge attachment' was answered in comment 3.

Main manuscript Page 12, Line 13: This difference could be attributed to the remaining lipid ligands that were attached to the corner site of the nanoparticles.

Comment 9: Supplementary Figure 5. The authors state in the caption of Figure 5 that Supplementary Figure Image 5-5 is used in Figure 1(a).

Response: We appreciate the reviewer's helpful comment. We re-measured TEM analysis of Ag-NP

samples for Figure 1a and supplementary Figure 1, 4, 5 and 6. All re-measured Ag-NP samples were sonicated in TMAH solution before drop-casted onto the TEM grids. And we modified the caption of Figure 1a in the revised manuscript. Please check the new Figure 1 and Supplementary Figure 7 on Page R3 and R8, respectively.

Main manuscript Page 5, Line 14: Fig. 1b: box plot and Supplementary Fig. 6: HR-TEM images for calculation of the particle size distribution; the image in Figure 1a was also used to estimate the particle size distribution.

REVIEWERS' COMMENTS

Reviewer #1 (Remarks to the Author):

In this revised manuscript, the authors have addressed the questions raised in the previous review. In this revision, recent results were added and further discussed, improving the manuscript's quality. Thus, the contribution could now be accepted for publication.

P.S. There is no scale bar in the Supplementary Figure 5.

Reviewer #3 (Remarks to the Author):

The authors have largely addressed my concerns in the current revision. I have just a few minor comments.

1. The authors should specify at the first mention of Ag black that it refers to the fine metal powder and also state in the methods the source of the Ag black powder.
2. Supplementary Figure 5 is missing the scale bar.
3. The main text will benefit from another round of proofreading by the authors. Some of the additions read somewhat awkwardly against the existing text.

REVIEWERS' COMMENTS

Reviewer #1 (Remarks to the Author):

In this revised manuscript, the authors have addressed the questions raised in the previous review. In this revision, recent results were added and further discussed, improving the manuscript's quality. Thus, the contribution could now be accepted for publication.

P.S. There is no scale bar in the Supplementary Figure 5.

Response: We appreciate the reviewer's helpful comment. We added scale bars in the Supplementary Figure 5.

Supplementary Figure 5. HR-TEM images of Ag-NP catalyst with 2 h of TMAH treatment for 2, 3, and 5-fold verification; scale bar = 3 nm.

Reviewer #3 (Remarks to the Author):

The authors have largely addressed my concerns in the current revision. I have just a few minor comments.

Comment 1: The authors should specify at the first mention of Ag black that it refers to the fine metal powder and also state in the methods the source of the Ag black powder.

Response: We appreciate the reviewer's helpful comment. In the revised manuscript, we added a phrase about Ag black catalyst that it refers to the fine powder of metallic Ag. Also we added a sentence about the source of Ag black catalyst.

Main manuscript Page 7, Line 5: Ag black catalyst is a commercial fine powder of metallic Ag.

Main manuscript Page 17, Line 21: A commercial Ag black catalyst was purchased from Alfa Aesar.

Comment 2: Supplementary Figure 5 is missing the scale bar.

Response: Thank you for this helpful comment. We added scale bars in the Supplementary Figure 5.

Supplementary Figure 5. HR-TEM images of Ag-NP catalyst with 2 h of TMAH treatment for 2, 3, and 5-fold verification; scale bar = 3 nm.

Comment 3: The main text will benefit from another round of proofreading by the authors. Some of the additions read somewhat awkwardly against the existing text.

Response: Thank you for bringing this issue to our attention. Proofreading was carried out on the phrases added in the last revision.